# Learning Feature Selection Dependencies in Multi-task Learning

**Daniel Hernández-Lobato**
Computer Science Department
Universidad Autónoma de Madrid
daniel.hernandez@uam.es

**José Miguel Hernández-Lobato**
Department of Engineering
University of Cambridge
jmh233@cam.ac.uk

## Abstract

A probabilistic model based on the horseshoe prior is proposed for learning dependencies in the process of identifying relevant features for prediction. Exact inference is intractable in this model. However, expectation propagation offers an approximate alternative. Because the process of estimating feature selection dependencies may suffer from over-fitting in the model proposed, additional data from a multi-task learning scenario are considered for induction. The same model can be used in this setting with few modifications. Furthermore, the assumptions made are less restrictive than in other multi-task methods: The different tasks must share feature selection dependencies, but can have different relevant features and model coefficients. Experiments with real and synthetic data show that this model performs better than other multi-task alternatives from the literature. The experiments also show that the model is able to induce suitable feature selection dependencies for the problems considered, only from the training data.

## 1 Introduction

Many linear regression problems are characterized by a large number $d$ of features or explaining attributes and by a reduced number $n$ of training instances. In this large $d$ but small $n$ scenario there is an infinite number of potential model coefficients that explain the training data perfectly well. To avoid over-fitting problems and to obtain estimates with good generalization properties, a typical regularization is to assume that the model coefficients are sparse, *i.e.*, most coefficients are equal to zero [1]. This is equivalent to considering that only a subset of the features or attributes are relevant for prediction. The sparsity assumption can be introduced by carrying out Bayesian inference under a sparsity enforcing prior for the model coefficients [2, 3], or by minimizing a loss function penalized by some sparse regularizer [4, 5]. Among the priors that enforce sparsity, the horseshoe has some attractive properties that are very convenient for the scenario described [3]. In particular, this prior has heavy tails, to model coefficients that significantly differ from zero, and an infinitely tall spike at the origin, to favor coefficients that take negligible values.

The estimation of the coefficients under the sparsity assumption can be improved by introducing dependencies in the process of determining which coefficients are zero [6, 7]. An extreme case of these dependencies appears in group feature selection methods in which groups of coefficients are considered to be jointly equal or different from zero [8, 9]. However, a practical limitation is that the dependency structure (groups) is often assumed to be given. Here, we propose a model based on the horseshoe prior that induces the dependencies in the feature selection process from the training data. These dependencies are expressed by a correlation matrix that is specified by $\mathcal{O}(d)$ parameters. Unfortunately, the estimation of these parameters from the training data is difficult since we consider $n < d$ instances only. Thus, over-fitting problems are likely to appear. To improve the estimation process we assume a multi-task learning setting, where several learning tasks share feature selection dependencies. The method proposed can be adapted to such a scenario with few modifications.

Traditionally, methods for multi-task learning under the sparsity assumption have considered common relevant and irrelevant features among tasks [8, 10, 11, 12, 13, 14]. Nevertheless, recent research cautions against this assumption when the supports and values of the coefficients for each task can vary widely [15]. The model proposed here limits the impact of this problem because it is has fewer restrictions. The tasks used for induction can have, besides different model coefficients, different relevant features. They must share only the dependency structure for the selection process.

The model described here is most related to the method for sparse coding introduced in [16], where spike-and-slab priors [2] are considered for multi-task linear regression under the sparsity assumption and dependencies in the feature selection process are specified by a Boltzmann machine. Fitting exactly the parameters of a Boltzmann machine to the observed data has exponential cost in the number of dimensions of the learning problem. Thus, when compared to the proposed model, the model considered in [16] is particularly difficult to train. For this, an approximate algorithm based on block-coordinate optimization has been described in [17]. The algorithm alternates between greedy MAP estimation of the sparsity patterns of each task and maximum pseudo-likelihood estimation of the Boltzmann parameters. Nevertheless, this algorithm lacks a proof of convergence and we have observed that is prone to get trapped in sub-optimal solutions.

Our experiments with real and synthetic data show the better performance of the model proposed when compared to other methods that try to overcome the problem of different supports among tasks. These methods include the model described in [16] and the model for dirty data proposed in [15]. These experiments also illustrate the benefits of the proposed model for inducing dependencies in the feature selection process. Specifically, the dependencies obtained are suitable for the multi-task learning problems considered. Finally, a difficulty of the model proposed is that exact Bayesian inference is intractable. Therefore, expectation propagation (EP) is employed for efficient approximate inference. In our model EP has a cost that is $\mathcal{O}(Kn^2d)$, where $K$ is the number of learning tasks, $n$ is the number of samples of each task, and $d$ is the dimensionality of the data.

The rest of the paper is organized as follows: Section 2 describes the proposed model for learning feature selection dependencies. Section 3 shows how to use expectation propagation to approximate the quantities required for induction. Section 4 compares this model with others from the literature on synthetic and real data regression problems. Finally, Section 5 gives the conclusions of the paper and some ideas for future work.

## 2    A Model for Learning Feature Selection Dependencies

We describe a linear regression model that can be used for learning dependencies in the process of identifying relevant features or attributes for prediction. For simplicity, we first deal with the case of a single learning task. Then, we show how this model can be extended to address multi-task learning problems. In the single task scenario we consider some training data in the form of $n$ $d$-dimensional vectors summarized in a design matrix $\mathbf{X} = (\mathbf{x}_1, \ldots, \mathbf{x}_n)^{\mathrm{T}}$ and associated targets $\mathbf{y} = (y_1, \ldots, y_n)^{\mathrm{T}}$, with $y_i \in \mathbb{R}$. A linear predictive rule is assumed for $\mathbf{y}$ given $\mathbf{X}$. Namely, $\mathbf{y} = \mathbf{X}\mathbf{w} + \boldsymbol{\epsilon}$, where $\mathbf{w}$ is a vector of latent coefficients and $\boldsymbol{\epsilon}$ is a vector of independent Gaussian noise with variance $\sigma^2$, *i.e.*, $\boldsymbol{\epsilon} \sim \mathcal{N}(\mathbf{0}, \sigma^2 \mathbf{I})$. Given $\mathbf{X}$ and $\mathbf{y}$, the likelihood for $\mathbf{w}$ is:

$$p(\mathbf{y}|\mathbf{X}, \mathbf{w}) = \prod_{i=1}^{n} p(y_i|\mathbf{x}_i, \mathbf{w}) = \prod_{i=1}^{n} \mathcal{N}(y_i|\mathbf{w}^{\mathrm{T}}\mathbf{x}_i, \sigma^2) = \mathcal{N}(\mathbf{y}|\mathbf{X}\mathbf{w}, \sigma^2 \mathbf{I}) . \qquad (1)$$

Consider the under-determined scenario $n < d$. In this case, the likelihood is not strictly concave and infinitely many values of $\mathbf{w}$ fit the training data perfectly well. A strong regularization technique that is often used in this context is to assume that only some features are relevant for prediction [1]. This is equivalent to assuming that $\mathbf{w}$ is sparse with many zeros. This inductive bias can be naturally incorporated into the model using a horseshoe sparsity enforcing prior for $\mathbf{w}$ [3].

The horseshoe prior lacks a closed form but can be defined as a scale mixture of Gaussians:

$$p(\mathbf{w}|\tau) = \prod_{j=1}^{d} p(w_j|\tau) , \qquad p(w_j|\tau) = \int \mathcal{N}(w_j|0, \lambda_j^2 \tau^2) \, \mathcal{C}^+(\lambda_j|0, 1) \, d\lambda_j , \qquad (2)$$

where $\lambda_j$ is a latent scale for coefficient $w_j$, $\mathcal{C}^+(\cdot|0, 1)$ is a half-Cauchy distribution with zero location and unit scale and $\tau > 0$ is a *global* shrinkage parameter that controls the level of sparsity. The

smaller the value of $\tau$ the sparser the prior and vice-versa. Figure 1 (left) and (middle) show a comparison of the horseshoe with other priors from the literature. The horseshoe has an infinitely tall spike at the origin which favors coefficients with small values, and has heavy tails which favor coefficients that take values that significantly differ from zero. Furthermore, assume that $\tau = \sigma^2 = 1$ and that $\mathbf{X} = \mathbf{I}$, and define $\kappa_j = 1/(1 + \lambda_j^2)$. Then, the posterior mean for $w_j$ is $(1 - \kappa_j)y_j$, where $\kappa_j$ is a random shrinkage coefficient that can be interpreted as the amount of weight placed at the origin [3]. Figure 1 (right) shows the prior density for $\kappa_j$ that results from the horseshoe. It is from the shape of this figure that the horseshoe takes its name. We note that one expects to see two things under this prior: relevant coefficients ($\kappa_j \approx 0$, no shrinkage), and zeros ($\kappa_j \approx 1$, total shrinkage). The horseshoe is therefore very convenient for the sparse inducing scenario described before.

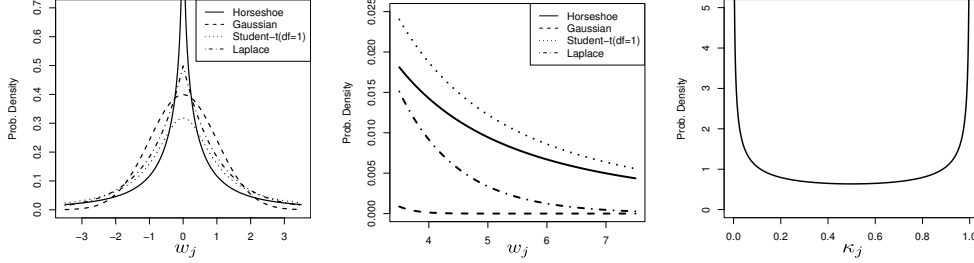

Figure 1: (left) Density of different priors, horseshoe, Gaussian, Student-t and Laplace near the origin. Note the infinitely tall spike of the horseshoe. (middle) Tails of the different priors considered before. (right) Prior density of the shrinkage parameter $\kappa_j$ for the horseshoe prior.

A limitation of the horseshoe is that it does not consider dependencies in the feature selection process. Specifically, the fact that one feature is actually relevant for prediction has no impact at all in the prior relevancy or irrelevancy of other features. We now describe how to introduce these dependencies in the horseshoe. Consider the definition of a Cauchy distribution as the ratio of two independent standard Gaussian random variables [18]. An equivalent representation of the prior is:

$$p(\mathbf{w}|\rho^2, \gamma^2) = \int \prod_{j=1}^{d} \mathcal{N}(w_j|0, u_j^2/v_j^2)\, \mathcal{N}(u_j|0, \rho^2)\, \mathcal{N}(v_j|0, \gamma^2)\, du_j dv_j\,. \tag{3}$$

where $u_j$ and $v_j$ are latent variables introduced for each dimension $j$. In particular, $\lambda_j = u_j\gamma/v_j\rho$. Furthermore, $\tau$ has been incorporated into the prior for $u_j$ and $v_j$ using $\tau^2 = \rho^2/\gamma^2$. The latent variables $u_j$ and $v_j$ can be interpreted as indicators of the relevance or irrelevance of feature $j$. The larger $u_j^2$, the more relevant the feature. Conversely, the larger $v_j^2$, the more irrelevant.

A simple way of introducing dependencies in the feature selection process is to consider correlations among variables $u_j$ and $v_j$, with $j = 1, \ldots, d$. These correlations can be introduced in (3) as follows:

$$p(\mathbf{w}|\rho^2, \gamma^2, \mathbf{C}) = \int \left[ \prod_{j=1}^{d} \mathcal{N}(w_j|0, u_j^2/v_j^2) \right] \mathcal{N}(\mathbf{u}|\mathbf{0}, \rho^2\mathbf{C})\, \mathcal{N}(\mathbf{v}|\mathbf{0}, \gamma^2\mathbf{C})\, d\mathbf{u}d\mathbf{v}\,, \tag{4}$$

where $\mathbf{u} = (u_1, \ldots, u_d)^{\mathrm{T}}$, $\mathbf{v} = (v_1, \ldots, v_d)^{\mathrm{T}}$, $\mathbf{C}$ is a correlation matrix that specifies the dependencies in the feature selection process, and $\rho^2$ and $\gamma^2$ act as regularization parameters that control the level of sparsity. When $\mathbf{C} = \mathbf{I}$, (4) factorizes and gives the same prior as the one in (2) and (3). In practice, however, $\mathbf{C}$ has to be estimated from the data. This can be problematic since it will involve the estimation of $\mathcal{O}(d^2)$ free parameters which can lead to over-fitting. To alleviate this problem and also to allow for efficient approximate inference we consider a special form for $\mathbf{C}$:

$$\mathbf{C} = \boldsymbol{\Delta}\mathbf{M}\boldsymbol{\Delta}\,, \qquad \mathbf{M} = (\mathbf{D} + \mathbf{P}\mathbf{P}^{\mathrm{T}})\,, \qquad \boldsymbol{\Delta} = \mathrm{diag}(1/\sqrt{M_{11}}, \ldots, 1/\sqrt{M_{dd}})\,, \tag{5}$$

where $\mathrm{diag}(a_1, \ldots, a_d)$ denotes a diagonal matrix with entries $a_1, \ldots, a_d$; $\mathbf{D}$ is a diagonal matrix whose entries are all equal to some small positive constant (this matrix guarantees that $\mathbf{C}^{-1}$ exists); the products by $\boldsymbol{\Delta}$ ensure that the entries of $\mathbf{C}$ are in the range $(-1, 1)$; and $\mathbf{P}$ is a $d \times m$ matrix of real entries which specifies the correlation structure of $\mathbf{C}$. Thus, $\mathbf{C}$ is fully determined by $\mathbf{P}$ and will only have $\mathcal{O}(md)$ free parameters with $m < d$. The value of $m$ is a regularization parameter that limits the complexity of $\mathbf{C}$. The larger its value, the more expressive $\mathbf{C}$ is. For computational reasons described later on we will set in our experiments $m$ equal to $n$, the number of data instances.

## 2.1 Inference, Prediction and Learning Feature Selection Dependencies

Denote by $\mathbf{z} = (\mathbf{w}^T, \mathbf{u}^T, \mathbf{v}^T)^T$ the vector of latent variables of the model described above. Based on the formulation of the previous section, the joint probability distribution of $\mathbf{y}$ and $\mathbf{z}$ is:

$$p(\mathbf{y}, \mathbf{z}|\mathbf{X}, \sigma^2, \rho^2, \gamma^2, \mathbf{C}) = \mathcal{N}(\mathbf{y}|\mathbf{Xw}, \sigma^2\mathbf{I})\mathcal{N}(\mathbf{u}|0, \rho^2\mathbf{C})\mathcal{N}(\mathbf{v}|0, \gamma^2\mathbf{C})\prod_{j=1}^{d}\mathcal{N}\left(w_j|0, u_j^2/v_j^2\right) . \quad (6)$$

Figure 2 shows the factor graph corresponding to this joint probability distribution. This graph summarizes the interactions between the random variables in the model. All the factors in (6) are Gaussian, except the ones corresponding to the prior for $w_j$ given $u_j$ and $v_j$, $\mathcal{N}(w_j|0, u_j^2/v_j^2)$.

Given the observed targets $\mathbf{y}$ one is typically interested in inferring the latent variables $\mathbf{z}$ of the model. For this, Bayes' theorem can be used:

$$p(\mathbf{z}|\mathbf{X}, \mathbf{y}, \sigma^2, \rho^2, \gamma^2, \mathbf{C}) = \frac{p(\mathbf{y}, \mathbf{z}|\mathbf{X}, \sigma^2, \rho^2, \gamma^2, \mathbf{C})}{p(\mathbf{y}|\mathbf{X}, \sigma^2, \rho^2, \gamma^2, \mathbf{C})} , \quad (7)$$

where the numerator in the r.h.s. of (7) is the joint distribution (6) and the denominator is simply a normalization constant (the model evidence) which can be used for Bayesian model selection [19].

The posterior distribution in (7) is useful to compute a predictive distribution for the target $y_{\text{new}}$ associated to a new unseen data instance $\mathbf{x}_{\text{new}}$:

$$p(y_{\text{new}}|\mathbf{x}_{\text{new}}, \mathbf{X}, \sigma^2, \rho^2, \gamma^2, \mathbf{C}) = \int p(y_{\text{new}}|\mathbf{x}_{\text{new}}, \mathbf{w})\, p(\mathbf{z}|\mathbf{X}, \sigma^2, \rho^2, \gamma^2, \mathbf{C})\, d\mathbf{z} . \quad (8)$$

Similarly, one can marginalize (7) with respect to $\mathbf{w}$ to obtain a posterior distribution for $\mathbf{u}$ and $\mathbf{v}$ which can be useful to identify the most relevant or irrelevant features.

Ideally, however, one should also infer $\mathbf{C}$, the correlation matrix that describes the dependencies in the feature selection process, and compute a posterior distribution for it. This can be complicated, even for approximate inference methods. Denote by $Z$ the model evidence, *i.e.*, the denominator in the r.h.s. of (7). A simpler alternative is to use gradient ascent to maximize $\log Z$ (and therefore $Z$) with respect to $\mathbf{P}$, the matrix that completely specifies $\mathbf{C}$. This corresponds to type-II maximum likelihood (ML) estimation and allows to determine $\mathbf{P}$ from the training data alone, without resorting to cross-validation [19]. The gradient of $\log Z$ with respect to $\mathbf{P}$, *i.e.*, $\partial \log Z/\partial\mathbf{P}$ can be used for this task. The other hyper-parameters of the model $\sigma^2$, $\rho^2$ and $\gamma^2$ can be found following a similar approach.

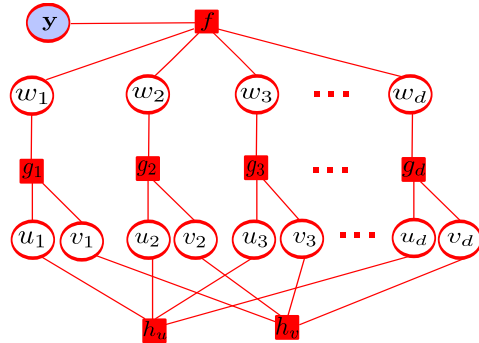

Figure 2: Factor graph of the probabilistic model. The factor $f(\cdot)$ corresponds to the likelihood $\mathcal{N}(y|\mathbf{Xw}, \sigma^2\mathbf{I})$, and each $g_j(\cdot)$ to the prior for $w_j$ given $u_j$ and $v_j$, $\mathcal{N}(w_j|0, u_j^2/v_j^2)$. Finally, $h_u(\cdot)$ and $h_v(\cdot)$ correspond to $\mathcal{N}(\mathbf{u}|\mathbf{0}, \rho^2\mathbf{C})$ and $\mathcal{N}(\mathbf{v}|\mathbf{0}, \gamma^2\mathbf{C})$, respectively. Only the targets $\mathbf{y}$ are observed, the other variables are latent.

Unfortunately, neither (7), (8) nor the model evidence can be computed in closed form. Specifically, it is not possible to compute the required integrals analytically. Thus, one has to resort to approximate inference. For this, we use expectation propagation [20]. See Section 3 for details.

## 2.2 Extension to the Multi-Task Learning Setting

In the single-task learning setting maximizing the model evidence with respect to $\mathbf{P}$ is not expected to be effective to improve the prediction accuracy. The reason is the difficulty of obtaining an accurate estimate of $\mathbf{P}$. This matrix has $m \times d$ free parameters and these have to be induced from a small number of $n < d$ training instances. The estimation process is hence likely to be affected by over-fitting. One way to mitigate over-fitting problems is to consider additional data for the estimation process. These additional data may come from a multi-task learning setting, where there are $K$

related but different tasks available for induction. A simple assumption is that all these tasks share a common dependency structure $\mathbf{C}$ for the feature selection process, although the model coefficients and the actual relevant features may differ between tasks. This assumption is less restrictive than assuming jointly relevant and irrelevant features across tasks and can be incorporated into the learning process using the described model with few modifications. By using the data from the $K$ tasks for the estimation of $\mathbf{P}$ we expect to obtain better estimates and to improve the prediction accuracy.

Assume there are $K$ learning tasks available for induction and that each task $k = 1, \ldots, K$ consists of a design matrix $\mathbf{X}_k$ with $n_k$ $d$-dimensional data instances and target values $\mathbf{y}_k$. As in (1), a linear predictive rule with additive Gaussian noise $\sigma_k^2$ is considered for each task. Let $\mathbf{w}_k$ be the model coefficients of task $k$. Assume for the model coefficients of each task a horseshoe prior as the one specified in (4) with a shared correlation matrix $\mathbf{C}$, but with task specific hyper-parameters $\rho_k^2$ and $\gamma_k^2$. Denote by $\mathbf{u}_k$ and $\mathbf{v}_k$ the vectors of latent Gaussian variables of the prior for task $k$. Similarly, let $\mathbf{z}_k = (\mathbf{w}_k^{\mathrm{T}}, \mathbf{u}_k^{\mathrm{T}}, \mathbf{v}_k^{\mathrm{T}})^{\mathrm{T}}$ be the vector of latent variables of task $k$. Then, the joint posterior distribution of the latent variables of the different tasks factorizes as follows:

$$p\left(\{\mathbf{z}\}_{k=1}^K | \{\mathbf{X}_k, \mathbf{y}_k, \tau_k^2, \rho_k^2, \sigma_k^2\}_{k=1}^K, \mathbf{C}\right) = \prod_{k=1}^K \frac{p(\mathbf{y}_k, \mathbf{z}_k | \mathbf{X}_k, \sigma_k^2, \rho_k^2, \gamma_k^2, \mathbf{C})}{p(\mathbf{y}_k | \mathbf{X}_k, \sigma_k^2, \rho_k^2, \gamma_k^2, \mathbf{C})}, \qquad (9)$$

where each factor in the r.h.s. of (9) is given by (7). This indicates that the $K$ models for each task can be learnt independently given $\mathbf{C}$ and $\sigma_k^2$, $\rho_k^2$ and $\gamma_k^2$ $\forall k$. Denote by $Z_{\mathrm{MT}}$ the denominator in the r.h.s. of (9), *i.e.*, $Z_{\mathrm{MT}} = \prod_{k=1}^K p(\mathbf{y}_k | \mathbf{X}_k, \sigma_k^2, \rho_k^2, \gamma_k^2, \mathbf{C}) = \prod_{k=1}^K Z_k$, with $Z_k$ the evidence for task $k$. Then, $Z_{\mathrm{MT}}$ is the model evidence for the multi-task setting. As in single-task learning, specific values for the hyper-parameters of each task and $\mathbf{C}$ can be found by a type-II maximum likelihood (ML) approach. For this, $\log Z_{\mathrm{MT}}$ is maximized using gradient ascent. Specifically, the gradient of $\log Z_{\mathrm{MT}}$ with respect to $\sigma_k^2$, $\rho_k^2$, $\gamma_k^2$ and $\mathbf{P}$ can be easily computed in terms of the gradient of each $\log Z_k$. In summary, if there is a method to approximate the required quantities for learning a single task using the model proposed, implementing a multi-task learning method that assumes shared feature selection dependencies but task dependent hyper-parameters is straight-forward.

## 3    Approximate Inference

Expectation propagation (EP) [20] is used to approximate the posterior distribution and the evidence of the model described in Section 2. For the clarity of presentation we focus on the model for a single learning task. The multi-task extension of Section 2.2 is straight-forward. Consider the posterior distribution of $\mathbf{z}$, (6). Up to a normalization constant this distribution can be written as

$$p(\mathbf{z}|\mathbf{X}, \mathbf{y}, \sigma^2, \rho^2, \gamma^2) \propto f(\mathbf{w}) h_u(\mathbf{u}) h_v(\mathbf{v}) \prod_{j=1}^d g_j(\mathbf{z}), \qquad (10)$$

where the factors in the r.h.s. of (10) are displayed in Figure 2. Note that all factors except the $g_j$'s are Gaussian. EP approximates (10) by a distribution $q(\mathbf{z}) \propto f(\mathbf{w}) h_u(\mathbf{u}) h_v(\mathbf{v}) \prod_{j=1}^d \tilde{g}_j(\mathbf{z})$, which is obtained by replacing each non-Gaussian factor $g_j$ in (10) with an approximate factor $\tilde{g}_j$ that is Gaussian but need not be normalized. Since the Gaussian distribution belongs to the exponential family of distributions, which is closed under the product and division operations [21], $q$ is Gaussian with natural parameters equal to the sum of the natural parameters of each factor.

EP iteratively updates each $\tilde{g}_j$ until convergence by first computing $q^{\setminus j} \propto q/\tilde{g}_j$ and then minimizing the Kullback-Leibler (KL) divergence between $g_j q^{\setminus j}$ and $q^{\mathrm{new}}$, $\mathrm{KL}(g_j q^{\setminus j} || q^{\mathrm{new}})$, with respect to $q^{\mathrm{new}}$. The new approximate factor is obtained as $\tilde{g}_j^{\mathrm{new}} = s_j q^{\mathrm{new}} / q^{\setminus j}$, where $s_j$ is the normalization constant of $g_j q^{\setminus j}$. This update rule ensures that $\tilde{g}_j$ looks similar to $g_j$ in regions of high posterior probability in terms of $q^{\setminus j}$ [20]. Minimizing the KL divergence is a convex problem whose optimum is found by matching the means and the covariance matrices between $g_j q^{\setminus j}$ and $q^{\mathrm{new}}$. These expectations can be readily obtained from the derivatives of $\log s_j$ with respect to the natural parameters of $q^{\setminus j}$ [21]. Unfortunately, the computation of $s_j$ is intractable under the horseshoe. As a practical alternative, our EP implementation employs numerical quadrature to evaluate $s_j$ and its derivatives. Importantly, $g_j$, and therefore $\tilde{g}_j$, depend only on $w_j$, $u_j$ and $v_j$, so a three-dimensional quadrature

will suffice. However, using similar arguments to those in [7] more efficient alternatives exist. Assume that $q^{\backslash j}(w_j, u_j, v_j) = \mathcal{N}(w_j|m_j, \eta_j)\mathcal{N}(u_j|0, \nu_j)\mathcal{N}(v_j|0, \xi_j)$, *i.e.*, $q^{\backslash j}$ factorizes with respect to $w_j$, $u_j$ and $v_j$ and that the mean of $u_j$ and $v_j$ is zero. Since $g_j$ is symmetric with respect to $u_j$ and $v_j$ then $\mathbb{E}[u_j] = \mathbb{E}[v_j] = \mathbb{E}[u_j v_j] = \mathbb{E}[u_j w_j] = \mathbb{E}[v_j w_j] = 0$ under $g_j q^{\backslash j}$. Thus, if the initial approximate factors $\tilde{g}_j$ factorize with respect to $w_j$, $u_j$ and $v_j$, and have zero mean with respect to $u_j$ and $v_j$, any updated factor will also satisfy these properties and $q^{\backslash j}$ will have the assumed form. The crucial point here is that the dependencies introduced by $g_j$ do not lead to correlations that need to be tracked under a Gaussian approximation. In this situation, the integral of $g_j q^{\backslash j}$ with respect to $w_j$ is given by the convolution of two Gaussians and the integral of the result with respect to $u_j$ and $v_j$ can be simplified using arguments similar to those employed to obtain (3). Namely,

$$s_j = \int \mathcal{N}\left(m_j|0, \frac{\nu_j}{\xi_j}\lambda_j^2 + \eta_j\right)\mathcal{C}^+(\lambda_j|0,1)d\lambda_j \,, \tag{11}$$

where $m_j$, $\eta_j$, $\nu_j$ and $\xi_j$ are the parameters of $q^{\backslash j}$. The derivatives of $\log s_j$ with respect to the natural parameters of $q^{\backslash j}$ can also be evaluated using a one-dimensional quadrature. Therefore, each update of $\tilde{g}_j$ requires five quadratures: one to evaluate $s_j$ and four to evaluate its derivatives.

Instead of sequentially updating each $\tilde{g}_j$, we follow [7] and update these factors in parallel. For this, we compute all $q^{\backslash j}$ at the same time and update each $\tilde{g}_j$. The marginals of $q$ are strictly required for this task. These can be efficiently obtained using the low rank representation structure of the covariance matrix of $q$ that results from the fact that all the $\tilde{g}_j$'s are factorizing univariate Gaussians and from the assumed form for $\mathbf{C}$ in (5). Specifically, if $m$ (the number of columns of $\mathbf{P}$) is equal to $n$, the cost of this operation (and hence the cost of EP) is $\mathcal{O}(n^2 d)$. Lastly, we damp the update of each $\tilde{g}_j$ as follows: $\tilde{g}_j = (\tilde{g}_j^{\text{new}})^\alpha (\tilde{g}_j^{\text{old}})^{1-\alpha}$, where $\tilde{g}_j^{\text{new}}$ and $\tilde{g}_j^{\text{old}}$ respectively denote the new and the old $\tilde{g}_j$, and $\alpha \in [0,1]$ is a parameter that controls the amount of damping. Damping significantly improves the convergence of EP and leaves the fixed points of the algorithm invariant [22].

After EP has converged, $q$ can be used instead of the exact posterior in (8) to make predictions. Similarly, the model evidence in (7) can be approximated by $\tilde{Z}$, the normalization constant of $q$:

$$\tilde{Z} = \int f(\mathbf{w})h_u(\mathbf{u})h_v(\mathbf{v})\prod_{j=1}^d \tilde{g}_j(\mathbf{z})d\mathbf{w}d\mathbf{u}d\mathbf{v} \,. \tag{12}$$

Since all the factors in (12) are Gaussian, $\log \tilde{Z}$ can be readily computed and maximized with respect to $\sigma^2$, $\rho^2$, $\gamma^2$ and $\mathbf{P}$ to find good values for these hyper-parameters. Specifically, once EP has converged, the gradient of the natural parameters of the $\tilde{g}_j$'s with respect to these hyper-parameters is zero [21]. Thus, the gradient of $\log \tilde{Z}$ with respect to $\sigma^2$, $\rho^2$, $\gamma^2$ and $\mathbf{P}$ can be computed in terms of the gradient of the exact factors. The derivations are long and tedious and hence omitted here, but by careful consideration of the covariance structure of $q$ it is possible to limit the complexity of these computations to $\mathcal{O}(n^2 d)$ if $m$ is equal to $n$. Therefore, to fit a model that maximizes $\log \tilde{Z}$ we alternate between running EP to obtain the estimate of $\log \tilde{Z}$ and its gradient, and doing a gradient ascent step to maximize this estimate with respect to $\sigma^2$, $\rho^2$, $\gamma^2$ and $\mathbf{P}$. The derivation details of the EP algorithm and an R-code implementation of it can be found in the supplementary material.

## 4 Experiments

We carry out experiments to evaluate the performance of the model described in Section 2. We refer to this model as $\text{HS}_{\text{Dep}}$. Other methods from the literature are also evaluated. The first one, $\text{HS}_{\text{ST}}$, is a particular case of $\text{HS}_{\text{Dep}}$ that is obtained when each task is learnt independently and correlations in the feature selection process are ignored (*i.e.*, $\mathbf{C} = \mathbf{I}$). A multi-task learning model, $\text{HS}_{\text{MT}}$, which assumes common relevant and irrelevant features among tasks is also considered. The details of this model are omitted, but it follows [10] closely. It assumes a horseshoe prior in which the scale parameters $\lambda_j$ in (2) are shared among tasks, *i.e.*, each feature is either relevant or irrelevant in *all* tasks. A variant of $\text{HS}_{MT}$, $\text{SS}_{\text{MT}}$, is also evaluated. $\text{SS}_{\text{MT}}$ considers a spike-and-slab prior for joint feature selection across all tasks, instead of a horseshoe prior. The details about the prior of $\text{SS}_{\text{MT}}$ are given in [10]. EP is used for approximate inference in both $\text{HS}_{\text{MT}}$ and $\text{SS}_{\text{MT}}$. The *dirty model*, DM, described in [15] is also considered. This model assumes shared relevant and irrelevant features

among tasks. However, some tasks are allowed to have specific relevant features. For this, a loss function is minimized via combined $\ell_1$ and $\ell_1/\ell_\infty$ block regularization. Particular cases of DM are the lasso [4] and the group lasso [8]. Finally, we evaluate the model introduced in [16]. This model, BM, uses spike-and-slab priors for feature selection and specifies dependencies in this process using a Boltzmann machine. BM is trained using the approximate block-coordinate algorithm described in [17]. All models considered assume Gaussian additive noise around the targets.

## 4.1 Experiments with Synthetic Data

A first batch of experiments is carried out using synthetic data. We generate $K = 64$ different tasks of $n = 64$ samples and $d = 128$ features. In each task, the entries of $\mathbf{X}_k$ are sampled from a standard Gaussian distribution and the model coefficients, $\mathbf{w}_k$, are all set to zero except for the $i$-th group of 8 consecutive coefficients, with $i$ chosen randomly for each task from the set $\{1, 2, \ldots, 16\}$. The values of these 8 non-zero coefficients are uniformly distributed in the interval $[-1, 1]$. Thus, in each task there are only 8 relevant features for prediction. Given each $\mathbf{X}_k$ and each $\mathbf{w}_k$, the targets $\mathbf{y}_k$ are obtained using (1) with $\sigma_k^2 = 0.5$ $\forall k$. The hyper-parameters of each method are set as follows: In HS$_{\text{ST}}$ $\rho_k^2$ and $\gamma_k^2$ are found by type-II ML. In HS$_{\text{MT}}$ $\rho^2$ and $\gamma^2$ are set to the average value found by HS$_{\text{ST}}$ for $\rho_k^2$ and $\gamma_k^2$, respectively. In SS$_{\text{MT}}$ the parameters of the spike-and-slab prior are found by type-II ML. In HS$_{\text{Dep}}$ $m = n$. Furthermore, $\rho_k^2$ and $\gamma_k^2$ take the values found by HS$_{\text{ST}}$ while $\mathbf{P}$ is obtained using type-II ML. In all models we set the variance of the noise for task $k$, $\sigma_k^2$, equal to $0.5$. Finally, in DM we try different hyper-parameters and report the best results observed. After training each model on the data, we measure the average reconstruction error of $\mathbf{w}_k$. Denote by $\hat{\mathbf{w}}_k$ the estimate of the model coefficients for task $k$ (this is the posterior mean except in BM and DM). The reconstruction error is measured as $||\hat{\mathbf{w}}_k - \mathbf{w}_k||_2 / ||\mathbf{w}_k||_2$, where $||\cdot||_2$ is the $\ell_2$-norm and $\mathbf{w}_k$ are the exact coefficients of task $k$.

| Method | Error |
|---|---|
| HS$_{\text{ST}}$ | 0.29±0.01 |
| HS$_{\text{MT}}$ | 0.38±0.03 |
| SS$_{\text{MT}}$ | 0.77±0.01 |
| DM | 0.37±0.01 |
| BM | 0.24±0.02 |
| HS$_{\text{Dep}}$ | 0.21±0.01 |

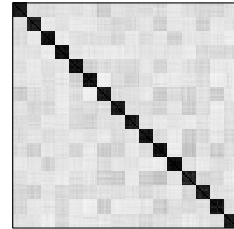

Figure 3 (top) shows the average reconstruction error of each method over 50 repetitions of the experiments described. HS$_{\text{Dep}}$ obtains the lowest error. The observed differences in performance are significant according to a Student's $t$-test (p-value $< 5\%$). BM performs worse than HS$_{\text{Dep}}$ because the greedy MAP estimation of the sparsity patterns of each task is sometimes trapped in sub-optimal solutions. The poor results of HS$_{\text{MT}}$, SS$_{\text{MT}}$ and DM are due to the assumption made by these models of *all* tasks sharing relevant features, which is not satisfied. Figure 3 (bottom) shows the average entries in absolute value of the correlation matrix $\mathbf{C}$ estimated by HS$_{\text{Dep}}$. The matrix has a block diagonal form, with blocks of size $8 \times 8$ (8 is the number of relevant coefficients in each task). Thus, within each block the corresponding latent variables $u_j$ and $v_j$ are strongly correlated, indicating jointly relevant or irrelevant features. This is the expected estimation for the scenario considered.

Figure 3: (top) Average reconstruction error of each method. (bottom) Average absolute value of the entries of the matrix $\mathbf{C}$ estimated by HS$_{\text{Dep}}$ in gray scale (white=0 and black=1). Black squares are groups of jointly relevant / irrelevant features.

## 4.2 Reconstruction of Images of Hand-written Digits from the MNIST

A second batch of experiments considers the reconstruction of images of hand-written digits extracted from the MNIST data set [23]. These images are in gray scale with pixel values between 0 and 255. Most pixels are inactive and equal to 0. Thus, the images are sparse and suitable to be reconstructed using the model proposed. The images are reduced to size $10 \times 10$ pixels and the pixel intensities are normalized to lie in the interval $[0, 1]$. Then, $K = 100$ tasks of $n = 75$ samples each are generated. For this, we randomly choose 50 images corresponding to the digit 3 and 50 images corresponding to the digit 5 (these digits are chosen because they differ significantly). Similar results (not shown) to the ones reported here are obtained for other pairs of digits. For each task, the entries of $\mathbf{X}_k$ are sampled from a standard Gaussian. The model coefficients, $\mathbf{w}_k$, are simply the pixel values of each image (*i.e.*, $d = 100$). Importantly, unlike in the previous experiments, the model coefficients are not synthetically generated but correspond to actual images. Furthermore, since the

tasks contain images of different digits they are expected to have different relevant features. Given $\mathbf{X}_k$ and $\mathbf{w}_k$, the targets $\mathbf{y}_k$ are generated using (1) with $\sigma_k^2 = 0.1 \ \forall k$. The objective is to reconstruct $\mathbf{w}_k$ from $\mathbf{X}_k$ and $\mathbf{y}_k$ for each task $k$. The hyper-parameters are set as in Section 4.1 with $\sigma_k^2 = 0.1$ $\forall k$. The reconstruction error is also measured as in that section.

Figure 4 (top) shows the average reconstruction error of each method over 50 repetitions of the experiments described. Again, $\text{HS}_{\text{Dep}}$ performs best. Furthermore, the differences in performance are also statistically significant. The second best result corresponds to $\text{HS}_{\text{MT}}$, probably due to background pixels which are irrelevant in *all* the tasks and to the heavy-tails of the horseshoe prior. $\text{HS}_{\text{ST}}$, $\text{SS}_{MT}$, BM and DM perform significantly worse. DM performs poorly probably because of the inferior shrinking properties of the $\ell_1$ norm compared to the horseshoe [3]. The poor results of $\text{SS}_{\text{MT}}$ are due to the lack of heavy-tails in the spike-and-slab prior. In BM we have observed that the greedy MAP estimation of the task supports is more frequently trapped in sub-optimal solutions. Furthermore, the algorithm described in [17] fails to converge most times in this scenario. Figure 4 (right, bottom) shows a representative subset of the images reconstructed by each method. The best reconstructions correspond to $\text{HS}_{\text{Dep}}$. Finally, Figure 4 (left, bottom) shows in gray scale the average correlations in absolute value induced by $\text{HS}_{\text{Dep}}$ for the selection process of each pixel of the image with respect to the selection of a particular pixel which is displayed in green. Correlations are high to avoid the selection of background pixels and to select pixels that actually correspond to the digits 3 and 5. The correlations induced are hence appropriate for the multi-task problem considered.

|  | $\text{HS}_{\text{ST}}$ | $\text{HS}_{\text{MT}}$ | $\text{SS}_{\text{MT}}$ | DM | BM | $\text{HS}_{\text{Dep}}$ |
|---|---|---|---|---|---|---|
| Error | 0.36±0.02 | 0.25±0.02 | 0.39±0.01 | 0.37±0.01 | 0.52±0.03 | 0.20±0.01 |

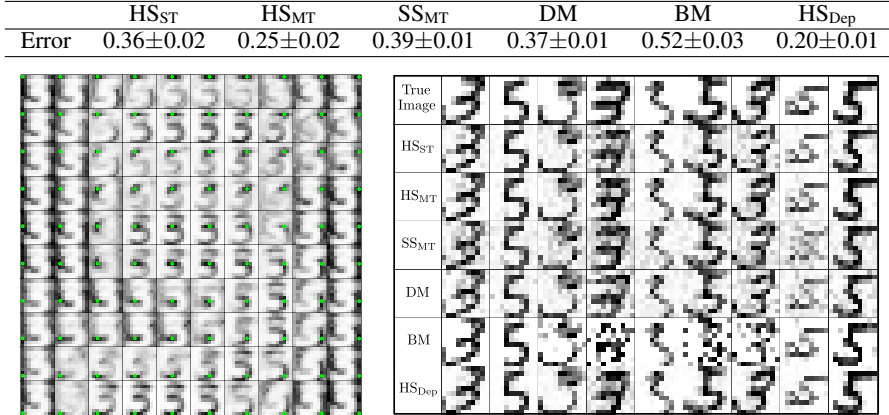

Figure 4: (top) Average reconstruction error each method. (left, bottom) Average absolute value correlation in a gray scale (white=0 and black=1) between the latent variables $u_j$ and $v_j$ corresponding to the pixel displayed in green and the variables $u_j$ and $v_j$ corresponding to all the other pixels of the image. (right, bottom) Examples of actual and reconstructed images by each method. The best reconstruction results correspond to $\text{HS}_{\text{Dep}}$.

## 5 Conclusions and Future Work

We have described a linear sparse model for learning dependencies in the feature selection process. The model can be used in a multi-task learning setting with several tasks available for induction that need not share relevant features, but only dependencies in the feature selection process. Exact inference is intractable in such a model. However, expectation propagation provides an efficient approximate alternative with a cost in $\mathcal{O}(Kn^2d)$, where $K$ is the number of tasks, $n$ is the number of samples of each task, and $d$ is the dimensionality of the data. Experiments with real and synthetic data illustrate the benefits of the proposed method. Specifically, this model performs better than other multi-task alternatives from the literature. Our experiments also show that the proposed model is able to induce relevant feature selection dependencies from the training data alone. Future paths of research include the evaluation of this model in practical problems of sparse coding, *i.e.*, when all tasks share a common design matrix $\mathbf{X}$ that has to be induced from the data alongside with the model coefficients, with potential applications to image denoising and image inpainting [24].

**Acknowledgment:** Daniel Hernández-Lobato is supported by the Spanish MCyT (Ref. TIN2010-21575-C02-02). José Miguel Hernández-Lobato is supported by Infosys Labs, Infosys Limited.

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
