[Supplementary Material · supplementary.pdf]

# Supplementary Material for: Learning Feature Selection Dependencies in Multi-task Learning

Daniel Hernández-Lobato
Computer Science Department
Universidad Autónoma de Madrid
Francisco Tomás y Valiente 11
28049, Madrid, Spain
daniel.hernandez@uam.es

José Miguel Hernández-Lobato
Department of Engineering
University of Cambridge
Trumpington Street, Cambridge
CB2 1PZ, United Kingdom
jmh233@eng.cam.ac.uk

## 1 EP Updates for the Horseshoe Prior

In this section we derive the EP updates of the approximate factors $\tilde{g}_j$. For simplicity, we focus exclusively on the updates without any damping effect. The inclusion of damping is straight-forward. It simply consists in setting for $j = 1, \ldots, d$, $\tilde{g}_j = \tilde{g}_{\text{new}}^\epsilon \tilde{g}_{\text{old}}^{1-\epsilon}$, where $\epsilon \in [0, 1]$ is the parameter that controls the amount of damping. In our experiments we set $\epsilon = 0.5$ and progressively decreased its value by 1% after each EP iteration.

We will consider the following form for the approximate factors $\tilde{g}_j$:

$$\tilde{g}_j(w_j, u_j, v_j) = \tilde{Z}_j \exp\left\{-\frac{1}{2\tilde{\eta}_j}(w_j - \tilde{m}_j)^2\right\} \exp\left\{-\frac{1}{2\tilde{\nu}_j}u_j^2\right\} \exp\left\{-\frac{1}{2\tilde{\xi}_j}v_j^2\right\} . \qquad (1)$$

That is, $\tilde{g}_j$ is a factorizing un-normalized Gaussian distribution with free parameters $\tilde{Z}_j$, $\tilde{m}_j$, $\tilde{\eta}_j$, $\tilde{\nu}_j$ and $\tilde{\xi}_j$ to be adjusted by EP.

The first step in the parallel EP algorithm is to compute the marginals of the current approximate posterior distribution. For this, the formulas for the product of Gaussian distributions have to be used. These formulas are found in the appendix of [1]. Denote by $q$ the posterior approximation, and by $\boldsymbol{\mu}$ and $\mathbf{V_w}$ the mean vector and covariance matrix of the Gaussian approximation for $\mathbf{w}$, respectively. Similarly, denote by $\mathbf{V_u}$ and $\mathbf{V_v}$ the covariance matrices of the Gaussian approximation for $\mathbf{u}$ and $\mathbf{v}$, respectively. Then, we have that $q(\mathbf{z}) = \mathcal{N}(\mathbf{w}|\boldsymbol{\mu}, \mathbf{V_w})\mathcal{N}(\mathbf{u}|\mathbf{0}, \mathbf{V_u})\mathcal{N}(\mathbf{v}|\mathbf{0}, \mathbf{V_v})$ and from the definition of $q(\mathbf{z}) \propto f(\mathbf{w})h_u(\mathbf{u})h_v(\mathbf{v})\prod_j \tilde{g}_j$ we have that

$$\mathbf{V_w}^{-1} = \boldsymbol{\Pi}_{\tilde{\eta}} + \frac{1}{\sigma^2}\mathbf{X}^{\mathrm{T}}\mathbf{X}, \qquad\qquad \boldsymbol{\mu} = \mathbf{V_w}\left(\frac{1}{\sigma^2}\mathbf{X}^{\mathrm{T}}\mathbf{y} + \boldsymbol{\Pi}_{\tilde{\eta}}\tilde{\mathbf{m}}\right),$$

$$\mathbf{V_u}^{-1} = \boldsymbol{\Pi}_{\tilde{\nu}} + \frac{1}{\rho^2}\mathbf{C}^{-1}, \qquad\qquad \mathbf{V_v}^{-1} = \boldsymbol{\Pi}_{\tilde{\xi}} + \frac{1}{\gamma^2}\mathbf{C}^{-1} \qquad (2)$$

where $\boldsymbol{\Pi}_{\tilde{\eta}}$, $\boldsymbol{\Pi}_{\tilde{\nu}}$ and $\boldsymbol{\Pi}_{\tilde{\xi}}$ are diagonal matrices whose $j$-th entries are equal to $\tilde{\eta}_j^{-1}$, $\tilde{\nu}_j^{-1}$ and $\tilde{\xi}_j^{-1}$, respectively, and where $\tilde{\mathbf{m}}$ is a vector whose $j$-th entry is equal to $\tilde{m}_j$. From these expressions $\mathbf{V_w}$, $\mathbf{V_u}$ and $\mathbf{V_v}$ can be efficiently computed using the Woodbury matrix identity formula. In particular,

$$\mathbf{V_w} = \boldsymbol{\Pi}_{\tilde{\eta}}^{-1} - \boldsymbol{\Pi}_{\tilde{\eta}}^{-1}\mathbf{X}^{\mathrm{T}}\left(\mathbf{I}\sigma^2 + \mathbf{X}\boldsymbol{\Pi}_{\tilde{\eta}}^{-1}\mathbf{X}^{\mathrm{T}}\right)^{-1}\mathbf{X}\boldsymbol{\Pi}_{\tilde{\eta}}^{-1} . \qquad (3)$$

$$(4)$$

Recall that $\mathbf{C} = \boldsymbol{\Delta}\mathbf{D}\boldsymbol{\Delta} + \boldsymbol{\Delta}\mathbf{P}\mathbf{P}^{\mathrm{T}}\boldsymbol{\Delta}$, with $\boldsymbol{\Delta}$ and $\mathbf{D}$ diagonal matrices. Then,

$$\mathbf{C}^{-1} = \boldsymbol{\Delta}^{-1}\mathbf{D}^{-1}\boldsymbol{\Delta}^{-1} - \boldsymbol{\Delta}^{-1}\mathbf{D}^{-1}\mathbf{P}\left(\mathbf{I} + \mathbf{P}^{\mathrm{T}}\mathbf{D}^{-1}\mathbf{P}\right)^{-1}\mathbf{P}^{\mathrm{T}}\mathbf{D}^{-1}\boldsymbol{\Delta}^{-1} \qquad (5)$$

and

$$\mathbf{V_u^{-1}} = \mathbf{\Pi}_{\tilde{\nu}} + \frac{1}{\rho^2}\mathbf{\Delta}^{-1}\mathbf{D}^{-1}\mathbf{\Delta}^{-1} - \mathbf{\Delta}^{-1}\mathbf{D}^{-1}\mathbf{P}\left(\mathbf{I}\rho^2 + \rho^2\mathbf{P}^{\mathrm{T}}\mathbf{D}^{-1}\mathbf{P}\right)^{-1}\mathbf{P}^{\mathrm{T}}\mathbf{D}^{-1}\mathbf{\Delta}^{-1}, \qquad (6)$$

$$\mathbf{V_v^{-1}} = \mathbf{\Pi}_{\tilde{\xi}} + \frac{1}{\gamma^2}\mathbf{\Delta}^{-1}\mathbf{D}^{-1}\mathbf{\Delta}^{-1} - \mathbf{\Delta}^{-1}\mathbf{D}^{-1}\mathbf{P}\left(\mathbf{I}\gamma^2 + \gamma^2\mathbf{P}^{\mathrm{T}}\mathbf{D}^{-1}\mathbf{P}\right)^{-1}\mathbf{P}^{\mathrm{T}}\mathbf{D}^{-1}\mathbf{\Delta}^{-1}, \qquad (7)$$

Define

$$\mathbf{G}_\rho = (\mathbf{\Pi}_{\tilde{\nu}} + \frac{1}{\rho^2}\mathbf{\Delta}^{-1}\mathbf{D}^{-1}\mathbf{\Delta}^{-1})^{-1} = \rho^2\mathbf{\Delta}\mathbf{D}\mathbf{\Delta}(\rho^2\mathbf{\Pi}_{\tilde{\nu}}\mathbf{\Delta}\mathbf{D}\mathbf{\Delta} + \mathbf{I})^{-1}, \qquad (8)$$

$$\mathbf{G}_\gamma = (\mathbf{\Pi}_{\tilde{\xi}} + \frac{1}{\gamma^2}\mathbf{\Delta}^{-1}\mathbf{D}^{-1}\mathbf{\Delta}^{-1})^{-1} = \gamma^2\mathbf{\Delta}\mathbf{D}\mathbf{\Delta}(\gamma^2\mathbf{\Pi}_{\tilde{\xi}}\mathbf{\Delta}\mathbf{D}\mathbf{\Delta} + \mathbf{I})^{-1}. \qquad (9)$$

Using again the Woodbury matrix identity formula we have that

$$\mathbf{V_u} = \mathbf{G}_\rho + \mathbf{G}_\rho\mathbf{\Delta}^{-1}\mathbf{D}^{-1}\mathbf{P}\left(\mathbf{I}\rho^2 + \rho^2\mathbf{P}^{\mathrm{T}}\mathbf{D}^{-1}\mathbf{P} - \mathbf{P}^{\mathrm{T}}\mathbf{D}^{-1}\mathbf{\Delta}^{-1}\mathbf{G}_\rho\mathbf{\Delta}^{-1}\mathbf{D}^{-1}\mathbf{P}\right)^{-1}\mathbf{P}^{\mathrm{T}}\mathbf{D}^{-1}\mathbf{\Delta}^{-1}\mathbf{G}_\rho$$
$$= \mathbf{G}_\rho + \mathbf{L}_\rho\left(\mathbf{I} + \mathbf{P}^{\mathrm{T}}\mathbf{B}_\rho\mathbf{P}\right)^{-1}\mathbf{L}_\rho^{\mathrm{T}}, \qquad (10)$$

$$\mathbf{V_v} = \mathbf{G}_\gamma + \mathbf{G}_\gamma\mathbf{\Delta}^{-1}\mathbf{D}^{-1}\mathbf{P}\left(\mathbf{I}\gamma^2 + \gamma^2\mathbf{P}^{\mathrm{T}}\mathbf{D}^{-1}\mathbf{P} - \mathbf{P}^{\mathrm{T}}\mathbf{D}^{-1}\mathbf{\Delta}^{-1}\mathbf{G}_\gamma\mathbf{\Delta}^{-1}\mathbf{D}^{-1}\mathbf{P}\right)^{-1}\mathbf{P}^{\mathrm{T}}\mathbf{D}^{-1}\mathbf{\Delta}^{-1}\mathbf{G}_\gamma$$
$$= \mathbf{G}_\gamma + \mathbf{L}_\gamma\left(\mathbf{I} + \mathbf{P}^{\mathrm{T}}\mathbf{B}_\gamma\mathbf{P}\right)^{-1}\mathbf{L}_\gamma^{\mathrm{T}}, \qquad (11)$$

where we have defined

$$\mathbf{L}_\rho = \rho\mathbf{\Delta}(\rho^2\mathbf{\Pi}_{\tilde{\nu}}\mathbf{\Delta}\mathbf{D}\mathbf{\Delta} + \mathbf{I})^{-1}\mathbf{P}, \qquad \mathbf{B}_\rho = \left(\rho^{-2}\mathbf{\Delta}^{-1}\mathbf{\Pi}_{\tilde{\nu}}^{-1}\mathbf{\Delta}^{-1} + \mathbf{D}\right)^{-1}, \qquad (12)$$

$$\mathbf{L}_\gamma = \gamma\mathbf{\Delta}(\rho^2\mathbf{\Pi}_{\tilde{\nu}}\mathbf{\Delta}\mathbf{D}\mathbf{\Delta} + \mathbf{I})^{-1}\mathbf{P}, \qquad \mathbf{B}_\gamma = \left(\gamma^{-2}\mathbf{\Delta}^{-1}\mathbf{\Pi}_{\tilde{\xi}}^{-1}\mathbf{\Delta}^{-1} + \mathbf{D}\right)^{-1}. \qquad (13)$$

$\mathbf{L}_\rho$ and $\mathbf{L}_\gamma$ are matrices of size $d \times m$. Similarly, $\left(\mathbf{I} + \mathbf{P}^{\mathrm{T}}\mathbf{B}_\gamma\mathbf{P}\right)$ and $\left(\mathbf{I} + \mathbf{P}^{\mathrm{T}}\mathbf{B}_\rho\mathbf{P}\right)$ are matrices of size $m \times m$. The consequence is that if $n < d$ and $m < d$ the diagonals of the matrices $\mathbf{V_w}$, $\mathbf{V_v}$ and $\mathbf{V_u}$, which are required to compute the marginal variances of $q$ can be obtained with cost $\mathcal{O}(n^2 d)$, $\mathcal{O}(m^2 d)$ and $\mathcal{O}(m^2 d)$, respectively. Computing the vector $\boldsymbol{\mu}$ has a cost $\mathcal{O}(n^2 d)$ using similar arguments.

Given the marginals of $q$, we compute for each $\tilde{g}_j$ the corresponding $q^{\backslash j}$ distribution as $q^{\backslash j} \propto q/\tilde{g}_j$, where all the latent variables that are different from $w_j$, $u_j$ and $v_j$ have been marginalized in $q$. Consider $q^{\backslash j}(w_j, v_j, u_j) = \mathcal{N}(w_j|m_j, \eta_j)\mathcal{N}(u_j|0, \nu_j)\mathcal{N}(v_j|0, \xi_j)$. Using the formulas for the quotient of Gaussian distributions, the parameters of $q^{\backslash j}$ are:

$$\eta_j = \left(V_{\mathbf{w}(j,j)}^{-1} - \tilde{\eta}_j^{-1}\right)^{-1}, \qquad\qquad m_j = \eta_j\left(V_{\mathbf{w}(j,j)}^{-1}\mu_j - \tilde{\eta}_j^{-1}\tilde{m}_j\right),$$

$$\nu_j = \left(V_{\mathbf{u}(j,j)}^{-1} - \tilde{\nu}_j^{-1}\right)^{-1}, \qquad\qquad \xi_j = \left(V_{\mathbf{v}(j,j)}^{-1} - \tilde{\xi}_j^{-1}\right)^{-1}. \qquad (14)$$

Given these parameters for each dimension $j$ we compute the optimal parameters of each approximate factor $\tilde{g}_j$ using the formulas described in the appendix of [1]. For this, we first compute the normalization constant $s_j$ of $g_j q^{\backslash j}$. This is done as follows:

$$s_j = \int_{-\infty}^{+\infty}\int_0^{+\infty}\mathcal{N}\left(w_j|0, \frac{\nu_j}{\xi_j}\lambda_j^2\right)\mathcal{C}^+\left(\lambda_j|0, 1\right)\mathcal{N}(w_j|m_j, \eta_j)d\lambda_j dw_j$$

$$= \int_0^{+\infty}\mathcal{N}\left(m_j|0, \frac{\nu_j}{\xi_j}\lambda_j^2 + \eta_j\right)\mathcal{C}^+\left(\lambda_j|0, 1\right)d\lambda_j$$

$$= \sqrt{\frac{\nu_j}{\xi_j}}\int_0^{+\infty}\mathcal{N}\left(m_j|0, \lambda_j^2 + \eta_j\right)\frac{2}{\pi\left(\frac{\nu_j}{\xi_j} + \lambda_j^2\right)}d\lambda_j, \qquad (15)$$

where we have used the convolution of Gaussians and $\mathcal{C}^+(\cdot|0, 1)$ is a truncated Cauchy distribution with zero location and unit scale. Furthermore, we have observed that the change of variables

performed provides more robust computations, since this integral, and also the following ones, are computed using quadrature techniques.

Next, we compute the expectation of $w_j$ under $g_j q^{\setminus j}$. This is done as follows:

$$\mathbb{E}_{g_j q^{\setminus j}}[w_j] = \frac{2}{s_j} \sqrt{\frac{\nu_j}{\xi_j}} \int_0^{+\infty} \mathcal{N}\left(m_j | 0, \lambda_j^2 + \eta_j\right) \frac{1}{\pi \left(\frac{\nu_j}{\xi_j} + \lambda_j^2\right)} \left[\left(\frac{1}{\lambda_j^2} + \frac{1}{\eta_j}\right)^{-1} \frac{m_j}{\eta_j}\right] d\lambda_j, \qquad (16)$$

where the right factor inside the integral is the mean of $w_j$ resulting from the following product $\mathcal{N}\left(w_j | 0, \frac{\nu_j}{\xi_j} \lambda_j^2\right) \mathcal{N}(w_j | m_j, \eta_j)$ for a fixed value of $\lambda_j$. The formulas for the product of Gaussians described in the appendix of [1] have been employed and we have performed the same change of variable as the one employed in the previous equation. When we do the marginzalization over $\lambda_j$, we obtain the exact mean.

The second moment of $w_j$ can be computed similarly. Namely,

$$\mathbb{E}_{g_j q^{\setminus j}}[w_j^2] = \frac{2}{s_j} \sqrt{\frac{\nu_j}{\xi_j}} \int_0^{+\infty} \frac{\mathcal{N}\left(m_j | 0, \lambda_j^2 + \eta_j\right)}{\pi \left(\frac{\nu_j}{\xi_j} + \lambda_j^2\right)} \left\{\left[\left(\frac{1}{\lambda_j^2} + \frac{1}{\eta_j}\right)^{-1} \frac{m_j}{\eta_j}\right]^2 + \left[\frac{1}{\lambda_j^2} + \frac{1}{\eta_j}\right]^{-1}\right\} d\lambda_j,$$
$$(17)$$

where the right factor inside the integral is the second moment of $w_j$ resulting from the following product $\mathcal{N}\left(w_j | 0, \frac{\nu_j}{\xi_j} \lambda_j^2\right) \mathcal{N}(w_j | m_j, \eta_j)$ for a fixed value of $\lambda_j$. Again, when we do the marginzalization over $\lambda_j$, we obtain the exact value of the moment.

Another possibility to obtain the first and second moments of $w_j$ under $g_j q^{\setminus j}$ is to compute the derivatives of $\log s_j$ with respect to $m_j$ and $\eta_j$ as indicated in the appendix of [1]. However, although they provide similar results, we have observed that the last two equations above give more robust computations.

For the computation of the variances of $v_j$ and $u_j$ under $g_j q^{\setminus j}$ we employ the gradient of $\log s_j$, with respect to $\nu_j$ and $\xi_j$, as indicated in the appendix of [1]. In particular, since $\mathbb{E}_{q_j q^{\setminus j}}[u_j] = \mathbb{E}_{q_j q^{\setminus j}}[v_j] = 0$ we have that

$$\mathbb{E}_{q_j q^{\setminus j}}[u_j^2] = \nu_j + 2\nu_j^2 \frac{\partial \log s_j}{\partial \nu_j} = \nu_j \left(1 + 2\nu_j \frac{\partial \log s_j}{\partial \nu_j}\right), \qquad (18)$$

$$\mathbb{E}_{q_j q^{\setminus j}}[v_j^2] = \xi_j + 2\xi_j^2 \frac{\partial \log s_j}{\partial \xi_j} = \xi_j \left(1 + 2\xi_j \frac{\partial \log s_j}{\partial \xi_j}\right). \qquad (19)$$

The gradients of $\partial \log s_j$ are then computed as:

$$\frac{\partial \log s_j}{\partial \nu_j} = \frac{1}{2}\frac{1}{\nu_j} + \frac{1}{s_j} \sqrt{\frac{\nu_j}{\xi_j}} \int_0^{+\infty} \mathcal{N}\left(m_j | 0, \lambda_j^2 + \eta_j\right) \frac{-2}{\left(\pi \left(\frac{\nu_j}{\xi_j} + \lambda_j^2\right)\right)^2} \frac{\pi}{\xi_j} d\lambda_j, \qquad (20)$$

$$\frac{\partial \log s_j}{\partial \xi_j} = -\frac{1}{2}\frac{1}{\xi_j} + \frac{1}{s_j} \sqrt{\frac{\nu_j}{\xi_j}} \int_0^{+\infty} \mathcal{N}\left(m_j | 0, \lambda_j^2 + \eta_j\right) \frac{2}{\left(\pi \left(\frac{\nu_j}{\xi_j} + \lambda_j^2\right)\right)^2} \pi \frac{\nu_j}{\xi_j^2} d\lambda_j. \qquad (21)$$

In consequence, $\xi_j \frac{\partial \log s_j}{\partial \xi_j} = -\nu_j \frac{\partial \log s_j}{\partial \nu_j}$ and only one quadrature is required to evaluate $\mathbb{E}_{q_j q^{\setminus j}}[u_j^2]$ and $\mathbb{E}_{q_j q^{\setminus j}}[v_j^2]$. Thus, in total only four quadratures are required instead of five to compute the moments of $g_j q^{\setminus j}$.

These moments are used to compute an updated distribution $q^{\text{new}}$. Then, we update the corresponding approximate factor $\tilde{g}_j$ by setting $\tilde{g}_j = s_j q^{\text{new}}/q^{\setminus j}$. The formulas for the ratio of Gaussian distributions have to be used. These formulas are found in the appendix of [1]. This

results in the following updates for the parameters of $\tilde{g}_j$:

$$\tilde{\eta}_j = \left( \left( \mathbb{E}_{g_j q^{\backslash j}}[w_j^2] - \mathbb{E}_{g_j q^{\backslash j}}[w_j]^2 \right)^{-1} - \eta_j^{-1} \right)^{-1}, \tag{22}$$

$$\tilde{m}_j = \tilde{\eta}_j \left( \left( \mathbb{E}_{g_j q^{\backslash j}}[w_j^2] - \mathbb{E}_{g_j q^{\backslash j}}[w_j]^2 \right)^{-1} \mathbb{E}_{g_j q^{\backslash j}}[w_j] - \eta_j^{-1} m_j \right), \tag{23}$$

$$\tilde{\nu}_j = \left( \left( \mathbb{E}_{g_j q^{\backslash j}}[u_j^2] \right)^{-1} - \nu_j^{-1} \right)^{-1}, \tag{24}$$

$$\tilde{\xi}_j = \left( \left( \mathbb{E}_{g_j q^{\backslash j}}[v_j^2] \right)^{-1} - \xi_j^{-1} \right)^{-1}, \tag{25}$$

$$\tilde{Z}_j = s_j \sqrt{\frac{\eta_j}{\mathbb{E}_{g_j q^{\backslash j}}[w_j^2] - \mathbb{E}_{g_j q^{\backslash j}}[w_j]^2}} \sqrt{\frac{\nu_j}{\mathbb{E}_{g_j q^{\backslash j}}[u_j^2]}} \sqrt{\frac{\xi_j}{\mathbb{E}_{g_j q^{\backslash j}}[v_j^2]}}$$

$$\exp\left\{ -\frac{1}{2} \left( \frac{\mathbb{E}_{g_j q^{\backslash j}}[w_j]^2}{\mathbb{E}_{g_j q^{\backslash j}}[w_j^2] - \mathbb{E}_{g_j q^{\backslash j}}[w_j]^2} - \frac{m_j^2}{\eta_j} - \frac{\tilde{m}_j^2}{\tilde{\eta}_j} \right) \right\}. \tag{26}$$

In our code we have further assumed that EP has converged for the computation of $\tilde{Z}_j$.

Once all approximate factors $\tilde{g}_j$ have been updated, we recompute $q$ as the product of all the factors, the exact ones and the approximate. For this, (2) and following derivations can be used.

## 2 EP Approximation of the Marginal Likelihood

Once EP has converged we evaluate the logarithm of the approximation to the marginal likelihood of the model, $\tilde{Z}$. This is done using the formulas for the product of Gaussians described in the appendix of [1]:

$$\log \tilde{Z} = \log \int f(\mathbf{w}) h_u(\mathbf{u}) h_v(\mathbf{v}) \prod_j \tilde{g}_j(\mathbf{z}) d\mathbf{z}$$

$$= \left[ \sum_j \log \tilde{Z}_j \right] - \frac{n}{2} \log(2\pi\sigma^2) - \frac{1}{2\sigma^2} \mathbf{y}^{\mathrm{T}} \mathbf{y} + \frac{d}{2} \log(2\pi) + \frac{1}{2} \log |\mathbf{V}_\mathbf{w}| + \frac{1}{2} \log |\mathbf{V}_\mathbf{u}| - \frac{1}{2} \log |\rho^2 \mathbf{C}|$$

$$+ \frac{1}{2} \log |\mathbf{V}_\mathbf{v}| - \frac{1}{2} \log |\gamma^2 \mathbf{C}| - \frac{1}{2} \tilde{\mathbf{m}}^{\mathrm{T}} \mathbf{\Pi}_{\tilde{\eta}} \tilde{\mathbf{m}} + \frac{1}{2} \boldsymbol{\mu}^{\mathrm{T}} \mathbf{V}_\mathbf{w}^{-1} \boldsymbol{\mu}, \tag{27}$$

where $\tilde{m}$ is a vector whose $j$-th component is equal to $\tilde{m}_j$, $n$ is the number of instances, $d$ is the number of features and $\sigma^2$ the variance of the Gaussian additive noise. We next describe how to efficiently compute some of the required quantities. In particular,

$$\log |\mathbf{V}_\mathbf{w}| = -\log \left| \mathbf{\Pi}_{\tilde{\eta}} + \frac{1}{\sigma^2} \mathbf{X}^{\mathrm{T}} \mathbf{X} \right| = -\log |\mathbf{\Pi}_{\tilde{\eta}}| - \log \left| \mathbf{I} + \frac{1}{\sigma^2} \mathbf{\Pi}_{\tilde{\eta}}^{-1} \mathbf{X}^{\mathrm{T}} \mathbf{X} \right|$$

$$= -\log |\mathbf{\Pi}_{\tilde{\eta}}| - \log \left| \mathbf{I} + \frac{1}{\sigma^2} \mathbf{X} \mathbf{\Pi}_{\tilde{\eta}}^{-1} \mathbf{X}^{\mathrm{T}} \right|, \tag{28}$$

$$\log |\mathbf{V}_\mathbf{u}| = \log \left| \mathbf{G}_\rho + \mathbf{L}_\rho \left( \mathbf{I} + \mathbf{P}^{\mathrm{T}} \mathbf{B}_\rho \mathbf{P} \right)^{-1} \mathbf{L}_\rho^{\mathrm{T}} \right|$$

$$= \log |\mathbf{G}_\rho| + \log \left| \mathbf{I} + \mathbf{G}_\rho^{-1} \mathbf{L}_\rho \left( \mathbf{I} + \mathbf{P}^{\mathrm{T}} \mathbf{B}_\rho \mathbf{P} \right)^{-1} \mathbf{L}_\rho^{\mathrm{T}} \right|$$

$$= \log |\mathbf{G}_\rho| + \log \left| \mathbf{I} + \left( \mathbf{I} + \mathbf{P}^{\mathrm{T}} \mathbf{B}_\rho \mathbf{P} \right)^{-1} \mathbf{L}_\rho^{\mathrm{T}} \mathbf{G}_\rho^{-1} \mathbf{L}_\rho \right|, \tag{29}$$

$$\log |\mathbf{V}_\mathbf{v}| = \log \left| \mathbf{G}_\gamma + \mathbf{L}_\gamma \left( \mathbf{I} + \mathbf{P}^{\mathrm{T}} \mathbf{B}_\gamma \mathbf{P} \right)^{-1} \mathbf{L}_\gamma^{\mathrm{T}} \right|$$

$$= \log |\mathbf{G}_\gamma| + \log \left| \mathbf{I} + \mathbf{G}_\gamma^{-1} \mathbf{L}_\gamma \left( \mathbf{I} + \mathbf{P}^{\mathrm{T}} \mathbf{B}_\gamma \mathbf{P} \right)^{-1} \mathbf{L}_\gamma^{\mathrm{T}} \right|$$

$$= \log |\mathbf{G}_\gamma| + \log \left| \mathbf{I} + \left( \mathbf{I} + \mathbf{P}^{\mathrm{T}} \mathbf{B}_\gamma \mathbf{P} \right)^{-1} \mathbf{L}_\gamma^{\mathrm{T}} \mathbf{G}_\gamma^{-1} \mathbf{L}_\gamma \right|, \tag{30}$$

where we have used Sylvester's determinant theorem and all computations have cost in $\mathcal{O}(n^2 d)$ if $m = n$. Furthermore,

$$
\begin{aligned}
\log|\rho^2 \mathbf{C}| &= d\log\rho^2 + \log|\mathbf{\Delta}\mathbf{D}\mathbf{\Delta} + \mathbf{\Delta}\mathbf{P}\mathbf{P}^{\mathrm{T}}\mathbf{\Delta}| = d\log\rho^2 + 2\log|\mathbf{\Delta}| + \log|\mathbf{D} + \mathbf{P}\mathbf{P}^{\mathrm{T}}| \\
&= d\log\rho^2 + 2\log|\mathbf{\Delta}| + \log|\mathbf{D}| + \log|\mathbf{I} + \mathbf{D}^{-1}\mathbf{P}\mathbf{P}^{\mathrm{T}}| \\
&= d\log\rho^2 + 2\log|\mathbf{\Delta}| + \log|\mathbf{D}| + \log|\mathbf{I} + \mathbf{P}^{\mathrm{T}}\mathbf{D}^{-1}\mathbf{P}|,
\end{aligned}
\tag{31}
$$

where again we have used Sylvester's determinant theorem. $\log|\gamma^2 \mathbf{C}|$ can be similarly computed. Finally,

$$
\begin{aligned}
\boldsymbol{\mu}^{\mathrm{T}}\mathbf{V}_{\mathbf{w}}^{-1}\boldsymbol{\mu} &= \boldsymbol{v}^{\mathrm{T}}\mathbf{V}_{\mathbf{w}}\boldsymbol{v} \\
&= \boldsymbol{v}^{\mathrm{T}}\mathbf{\Pi}_{\tilde{\eta}}^{-1}\boldsymbol{v} - \boldsymbol{v}^{\mathrm{T}}\mathbf{\Pi}_{\tilde{\eta}}^{-1}\mathbf{X}^{\mathrm{T}}\left(\mathbf{I}\sigma^2 + \mathbf{X}\mathbf{\Pi}_{\tilde{\eta}}^{-1}\mathbf{X}^{\mathrm{T}}\right)^{-1}\mathbf{X}\mathbf{\Pi}_{\tilde{\eta}}^{-1}\boldsymbol{v},
\end{aligned}
\tag{32}
$$

where we have defined $\boldsymbol{v} = \left(\frac{1}{\sigma^2}\mathbf{X}^{\mathrm{T}}\mathbf{y} + \mathbf{\Pi}_{\tilde{\eta}}\tilde{\mathbf{m}}\right)$.

# 3   Computation of the Gradients of the Approximation

In this section we compute the gradient of $\log\tilde{Z}$ with respect to the different hyper-parameters of the model, *i.e.*, $\sigma^2$, $\rho^2$, $\gamma^2$ and $\mathbf{P}$. This is done as follows:

$$
\frac{\partial \log\tilde{Z}}{\partial\sigma^2} = -\frac{n}{2}\frac{1}{\sigma^2} + \frac{1}{2\sigma^4}\mathbf{y}^{\mathrm{T}}\mathbf{y} + \frac{1}{2}\frac{\partial\log|\mathbf{V}_w|}{\partial\sigma^2} + \frac{1}{2}\frac{\partial}{\partial\sigma^2}\boldsymbol{\mu}^{\mathrm{T}}\mathbf{V}_{\mathbf{w}}^{-1}\boldsymbol{\mu},
\tag{33}
$$

where we have that

$$
\frac{\partial\log|\mathbf{V}_{\mathbf{w}}|}{\partial\sigma^2} = -\mathrm{tr}\left(\mathbf{V}_{\mathbf{w}}\frac{\partial\mathbf{V}_{\mathbf{w}}^{-1}}{\partial\sigma^2}\right) = \mathrm{tr}\left(\mathbf{V}_{\mathbf{w}}\frac{1}{\sigma^4}\mathbf{X}^{\mathrm{T}}\mathbf{X}\right) = \frac{1}{\sigma^4}\mathrm{tr}\left(\mathbf{V}_{\mathbf{w}}\mathbf{X}^{\mathrm{T}}\mathbf{X}\right) = \frac{1}{\sigma^4}\mathrm{tr}\left(\mathbf{X}\mathbf{V}_{\mathbf{w}}\mathbf{X}^{\mathrm{T}}\right),
$$

$$
\frac{\partial}{\partial\sigma^2}\boldsymbol{\mu}^{\mathrm{T}}\mathbf{V}_{\mathbf{w}}^{-1}\boldsymbol{\mu} = \frac{\partial}{\partial\sigma^2}\boldsymbol{v}^{\mathrm{T}}\mathbf{V}_{\mathbf{w}}\boldsymbol{v} = 2\boldsymbol{v}^{\mathrm{T}}\mathbf{V}_{\mathbf{w}}\left(\frac{\partial}{\partial\sigma^2}\boldsymbol{v}\right) + \boldsymbol{v}^{\mathrm{T}}\left(\frac{\partial}{\partial\sigma^2}\mathbf{V}_{\mathbf{w}}\right)\boldsymbol{v}.
\tag{34}
$$

which can be easily computed in $\mathcal{O}(n^2 d)$ steps using the special representation for $\mathbf{V}_{\mathbf{w}}$ described at the beginning of this document, if $n \ll d$. In particular,

$$
\frac{\partial\boldsymbol{v}}{\partial\sigma^2} = -\frac{1}{\sigma^4}\mathbf{X}^{\mathrm{T}}\mathbf{y}, \qquad\qquad \frac{\partial\mathbf{V}_{\mathbf{w}}}{\partial\sigma^2} = \mathbf{\Pi}_{\tilde{\eta}}^{-1}\mathbf{X}^{\mathrm{T}}\mathbf{M}^{-1}\mathbf{M}^{-1}\mathbf{X}\mathbf{\Pi}_{\tilde{\eta}}^{-1},
\tag{35}
$$

where we have defined $\mathbf{M} = \left(\mathbf{I}\sigma^2 + \mathbf{X}\mathbf{\Pi}_{\tilde{\eta}}^{-1}\mathbf{X}^{\mathrm{T}}\right)$. Thus, the gradient with respect to $\sigma^2$ can be computed in $\mathcal{O}(n^2 d)$ steps, under the assumption that $m = n$ and $n \ll d$.

We now compute the gradient with respect to $\rho^2$ and $\gamma^2$. That is,

$$
\frac{\partial\log\tilde{Z}}{\partial\rho^2} = \frac{1}{2}\frac{\partial\log|\mathbf{V}_{\mathbf{u}}|}{\partial\rho^2} - \frac{1}{2}\frac{d}{\rho^2},
\tag{36}
$$

where we have that

$$
\begin{aligned}
\frac{\partial\log|\mathbf{V}_{\mathbf{u}}|}{\partial\rho^2} &= -\mathrm{tr}\left(\mathbf{V}_{\mathbf{u}}\frac{\partial\mathbf{V}_{\mathbf{u}}^{-1}}{\partial\rho^2}\right) = \mathrm{tr}\left(\mathbf{V}_{\mathbf{u}}\frac{1}{\rho^4}\mathbf{C}^{-1}\right) = \frac{1}{\rho^4}\mathrm{tr}\left(\mathbf{V}_{\mathbf{u}}\mathbf{C}^{-1}\right) = \frac{1}{\rho^2}\mathrm{tr}\left(\left(\mathbf{C}\mathbf{\Pi}_{\tilde{\nu}}\rho^2 + \mathbf{I}\right)^{-1}\right) \\
&= \frac{1}{\rho^2}\mathrm{tr}\left(\mathbf{I} - \rho^2\left(\mathbf{C}^{-1} + \rho^2\mathbf{\Pi}_{\tilde{\nu}}\right)^{-1}\mathbf{\Pi}_{\tilde{\nu}}\right) = \frac{1}{\rho^2}\mathrm{tr}\left(\mathbf{I} - \left(\mathbf{C}^{-1}\rho^{-2} + \mathbf{\Pi}_{\tilde{\nu}}\right)^{-1}\mathbf{\Pi}_{\tilde{\nu}}\right) \\
&= \frac{1}{\rho^2}\mathrm{tr}\left(\mathbf{I} - \mathbf{V}_{\mathbf{u}}\mathbf{\Pi}_{\tilde{\nu}}\right) = \frac{d}{2}\frac{1}{\rho^2} - \frac{1}{\rho^2}\mathrm{tr}\left(\mathbf{V}_{\mathbf{u}}\mathbf{\Pi}_{\tilde{\nu}}\right).
\end{aligned}
\tag{37}
$$

In the last expression we have used the Woodbury formula and the definition of $\mathbf{V}_{\mathbf{u}}$. The consequence is that

$$
\frac{\partial\log\tilde{Z}}{\partial\rho^2} = \frac{1}{2\rho^2}\mathrm{tr}\left(\mathbf{V}_{\mathbf{u}}\mathbf{\Pi}_{\tilde{\nu}}\right),
\tag{38}
$$

which can be easily computed in $\mathcal{O}(n^2 d)$ steps if $m = n$ and $n \ll d$. Specifically, for this we only need the marginal posterior variances of $\mathbf{u}$.

The gradient with respect to $\gamma^2$ can be computed similarly. The result is:

$$\frac{\partial \log \tilde{Z}}{\partial \gamma^2} = \frac{1}{2\gamma^2} \text{tr} \left( \mathbf{V_v} \mathbf{\Pi}_{\tilde{\xi}} \right) . \tag{39}$$

Finally, we compute the gradient with respect to each entry of $\mathbf{P}$, $P_{i,j}$. This is done as follows:

$$\frac{\partial \log \tilde{Z}}{\partial P_{i,j}} = \frac{1}{2} \frac{\partial \log |\mathbf{V_u}|}{\partial P_{i,j}} - \frac{1}{2} \frac{\partial \log |\rho^2 \mathbf{C}|}{\partial P_{i,j}} + \frac{1}{2} \frac{\partial \log |\mathbf{V_v}|}{\partial P_{i,j}} - \frac{1}{2} \frac{\partial \log |\gamma^2 \mathbf{C}|}{\partial P_{i,j}} , \tag{40}$$

where we have that

$$\frac{\partial \log |\mathbf{V_u}|}{\partial P_{i,j}} = -\text{tr} \left( \mathbf{V_u} \frac{\partial \mathbf{V_u^{-1}}}{\partial P_{i,j}} \right) = \text{tr} \left( \mathbf{V_u} \frac{1}{\rho^2} \mathbf{C}^{-1} \frac{\partial \mathbf{C}}{\partial P_{i,j}} \mathbf{C}^{-1} \right) . \tag{41}$$

Furthermore,

$$\frac{\partial \mathbf{C}}{\partial P_{i,j}} = 2\mathbf{\Delta D} \frac{\partial \mathbf{\Delta}}{\partial P_{i,j}} + \frac{\partial \mathbf{\Delta}}{\partial P_{i,j}} \mathbf{P} \mathbf{P}^{\mathrm{T}} \mathbf{\Delta} + \mathbf{\Delta} \mathbf{P} \mathbf{P}^{\mathrm{T}} \frac{\partial \mathbf{\Delta}}{\partial P_{i,j}} + \mathbf{\Delta} \boldsymbol{\delta}_i \boldsymbol{\delta}_j^{\mathrm{T}} \mathbf{P}^{\mathrm{T}} \mathbf{\Delta} + \mathbf{\Delta} \mathbf{P} \boldsymbol{\delta}_j \boldsymbol{\delta}_i^{\mathrm{T}} \mathbf{\Delta} , \tag{42}$$

where $\boldsymbol{\delta}_i$ and $\boldsymbol{\delta}_j$ are two vectors of sizes $d$ and $m$, respectively with all components equal to zero, except for components $i$-th and $j$-th, respectively, which are equal to one. In addition,

$$\frac{\partial \mathbf{\Delta}}{\partial P_{i,j}} = -\boldsymbol{\delta}_i \boldsymbol{\delta}_i^{\mathrm{T}} \mathbf{\Delta} \mathbf{\Delta} \mathbf{\Delta} \mathbf{P} \boldsymbol{\delta}_j \boldsymbol{\delta}_j^{\mathrm{T}} \boldsymbol{\delta}_j \boldsymbol{\delta}_i^{\mathrm{T}} . \tag{43}$$

Thus, the previous gradient is:

$$\begin{aligned}
\frac{\partial \log |\mathbf{V_u}|}{\partial P_{i,j}} &= -2\text{tr} \left( \mathbf{V_u} \frac{1}{\rho^2} \mathbf{C}^{-1} \mathbf{\Delta D} \boldsymbol{\delta}_i \boldsymbol{\delta}_i^{\mathrm{T}} \mathbf{\Delta} \mathbf{\Delta} \mathbf{\Delta} \mathbf{P} \boldsymbol{\delta}_j \boldsymbol{\delta}_j^{\mathrm{T}} \boldsymbol{\delta}_j \boldsymbol{\delta}_i^{\mathrm{T}} \mathbf{C}^{-1} \right) \\
&\quad - \text{tr} \left( \mathbf{V_u} \frac{1}{\rho^2} \mathbf{C}^{-1} \boldsymbol{\delta}_i \boldsymbol{\delta}_i^{\mathrm{T}} \mathbf{\Delta} \mathbf{\Delta} \mathbf{\Delta} \mathbf{P} \boldsymbol{\delta}_j \boldsymbol{\delta}_j^{\mathrm{T}} \boldsymbol{\delta}_j \boldsymbol{\delta}_i^{\mathrm{T}} \mathbf{P} \mathbf{P}^{\mathrm{T}} \mathbf{\Delta} \mathbf{C}^{-1} \right) \\
&\quad - \text{tr} \left( \mathbf{V_u} \frac{1}{\rho^2} \mathbf{C}^{-1} \mathbf{\Delta} \mathbf{P} \mathbf{P}^{\mathrm{T}} \boldsymbol{\delta}_i \boldsymbol{\delta}_i^{\mathrm{T}} \mathbf{\Delta} \mathbf{\Delta} \mathbf{\Delta} \mathbf{P} \boldsymbol{\delta}_j \boldsymbol{\delta}_j^{\mathrm{T}} \boldsymbol{\delta}_j \boldsymbol{\delta}_i^{\mathrm{T}} \mathbf{C}^{-1} \right) \\
&\quad + \text{tr} \left( \mathbf{V_u} \frac{1}{\rho^2} \mathbf{C}^{-1} \mathbf{\Delta} \boldsymbol{\delta}_i \boldsymbol{\delta}_j^{\mathrm{T}} \mathbf{P}^{\mathrm{T}} \mathbf{\Delta} \mathbf{C}^{-1} \right) \\
&\quad + \text{tr} \left( \mathbf{V_u} \frac{1}{\rho^2} \mathbf{C}^{-1} \mathbf{\Delta} \mathbf{P} \boldsymbol{\delta}_j \boldsymbol{\delta}_i^{\mathrm{T}} \mathbf{\Delta} \mathbf{C}^{-1} \right) \\
&= -2\text{tr} \left( \mathbf{V_u} \frac{1}{\rho^2} \mathbf{C}^{-1} \mathbf{\Delta D} \boldsymbol{\delta}_i \boldsymbol{\delta}_i^{\mathrm{T}} \mathbf{\Delta} \mathbf{\Delta} \mathbf{\Delta} \mathbf{P} \boldsymbol{\delta}_j \boldsymbol{\delta}_j^{\mathrm{T}} \boldsymbol{\delta}_j \boldsymbol{\delta}_i^{\mathrm{T}} \mathbf{C}^{-1} \right) \\
&\quad - 2\text{tr} \left( \mathbf{V_u} \frac{1}{\rho^2} \mathbf{C}^{-1} \boldsymbol{\delta}_i \boldsymbol{\delta}_i^{\mathrm{T}} \mathbf{\Delta} \mathbf{\Delta} \mathbf{\Delta} \mathbf{P} \boldsymbol{\delta}_j \boldsymbol{\delta}_j^{\mathrm{T}} \boldsymbol{\delta}_j \boldsymbol{\delta}_i^{\mathrm{T}} \mathbf{P} \mathbf{P}^{\mathrm{T}} \mathbf{\Delta} \mathbf{C}^{-1} \right) \\
&\quad + 2\text{tr} \left( \mathbf{V_u} \frac{1}{\rho^2} \mathbf{C}^{-1} \mathbf{\Delta} \boldsymbol{\delta}_i \boldsymbol{\delta}_j^{\mathrm{T}} \mathbf{P}^{\mathrm{T}} \mathbf{\Delta} \mathbf{C}^{-1} \right) \\
&= -2\text{tr} \left( \boldsymbol{\delta}_j^{\mathrm{T}} \boldsymbol{\delta}_j \boldsymbol{\delta}_i^{\mathrm{T}} \mathbf{C}^{-1} \mathbf{V_u} \frac{1}{\rho^2} \mathbf{C}^{-1} \mathbf{\Delta D} \boldsymbol{\delta}_i \boldsymbol{\delta}_i^{\mathrm{T}} \mathbf{\Delta} \mathbf{\Delta} \mathbf{\Delta} \mathbf{P} \boldsymbol{\delta}_j \right) \\
&\quad - 2\text{tr} \left( \boldsymbol{\delta}_j^{\mathrm{T}} \boldsymbol{\delta}_j \boldsymbol{\delta}_i^{\mathrm{T}} \mathbf{P} \mathbf{P}^{\mathrm{T}} \mathbf{\Delta} \mathbf{C}^{-1} \mathbf{V_u} \frac{1}{\rho^2} \mathbf{C}^{-1} \boldsymbol{\delta}_i \boldsymbol{\delta}_i^{\mathrm{T}} \mathbf{\Delta} \mathbf{\Delta} \mathbf{\Delta} \mathbf{P} \boldsymbol{\delta}_j \right) \\
&\quad + 2\text{tr} \left( \boldsymbol{\delta}_j^{\mathrm{T}} \mathbf{P}^{\mathrm{T}} \mathbf{\Delta} \mathbf{C}^{-1} \mathbf{V_u} \frac{1}{\rho^2} \mathbf{C}^{-1} \mathbf{\Delta} \boldsymbol{\delta}_i \right) .
\end{aligned} \tag{44}$$

In addition we have that:

$$\frac{\partial \log |\rho^2 \mathbf{C}|}{\partial P_{i,j}} = \frac{\partial \log |\mathbf{C}|}{\partial P_{i,j}} = \text{tr}\left(\mathbf{C}^{-1}\frac{\partial \mathbf{C}}{\partial P_{i,j}}\right)$$

$$= -2\text{tr}\left(\boldsymbol{\delta}_j^{\text{T}}\boldsymbol{\delta}_j\boldsymbol{\delta}_i^{\text{T}}\mathbf{C}^{-1}\boldsymbol{\Delta}\mathbf{D}\boldsymbol{\delta}_i\boldsymbol{\delta}_i^{\text{T}}\boldsymbol{\Delta}\boldsymbol{\Delta}\boldsymbol{\Delta}\mathbf{P}\boldsymbol{\delta}_j\right)$$

$$- 2\text{tr}\left(\boldsymbol{\delta}_j^{\text{T}}\boldsymbol{\delta}_j\boldsymbol{\delta}_i^{\text{T}}\mathbf{P}\mathbf{P}^{\text{T}}\boldsymbol{\Delta}\mathbf{C}^{-1}\boldsymbol{\delta}_i\boldsymbol{\delta}_i^{\text{T}}\boldsymbol{\Delta}\boldsymbol{\Delta}\boldsymbol{\Delta}\mathbf{P}\boldsymbol{\delta}_j\right)$$

$$+ 2\text{tr}\left(\boldsymbol{\delta}_j^{\text{T}}\mathbf{P}^{\text{T}}\boldsymbol{\Delta}\mathbf{C}^{-1}\boldsymbol{\Delta}\boldsymbol{\delta}_i\right) . \tag{45}$$

Furthermore, we can write

$$\left(\mathbf{C}^{-1}\frac{1}{\rho^2}\mathbf{V_u}\mathbf{C}^{-1} - \mathbf{C}^{-1}\right) = \left(\mathbf{C}^{-1}\frac{1}{\rho^2}\left(\boldsymbol{\Pi}_{\tilde{\nu}} + \frac{1}{\rho^2}\mathbf{C}^{-1}\right)^{-1}\mathbf{C}^{-1} - \mathbf{C}^{-1}\right)$$

$$= \left(\mathbf{C}^{-1}\left(\rho^2\boldsymbol{\Pi}_{\tilde{\nu}} + \mathbf{C}^{-1}\right)^{-1}\mathbf{C}^{-1} - \mathbf{C}^{-1}\right)$$

$$= -\left(\mathbf{C} + \frac{1}{\rho^2}\boldsymbol{\Pi}_{\tilde{\nu}}^{-1}\right)^{-1}$$

$$= -\boldsymbol{\Pi}_{\tilde{\nu}}\left(\boldsymbol{\Pi}_{\tilde{\nu}}\mathbf{C}\boldsymbol{\Pi}_{\tilde{\nu}} + \frac{1}{\rho^2}\boldsymbol{\Pi}_{\tilde{\nu}}\right)^{-1}\boldsymbol{\Pi}_{\tilde{\nu}}$$

$$= -\boldsymbol{\Pi}_{\tilde{\nu}}\left(\rho^2\boldsymbol{\Pi}_{\tilde{\nu}}^{-1} - \rho^2\left(\mathbf{C}^{-1} + \rho^2\boldsymbol{\Pi}_{\tilde{\nu}}\right)^{-1}\rho^2\right)\boldsymbol{\Pi}_{\tilde{\nu}}$$

$$= -\left(\rho^2\boldsymbol{\Pi}_{\tilde{\nu}} - \rho^2\boldsymbol{\Pi}_{\tilde{\nu}}\left(\mathbf{C}^{-1}\frac{1}{\rho^2} + \boldsymbol{\Pi}_{\tilde{\nu}}\right)^{-1}\boldsymbol{\Pi}_{\tilde{\nu}}\right)$$

$$= -\rho^2\left(\boldsymbol{\Pi}_{\tilde{\nu}} - \boldsymbol{\Pi}_{\tilde{\nu}}\mathbf{V_u}\boldsymbol{\Pi}_{\tilde{\nu}}\right) \tag{46}$$

where we have employed several times the Woodbury formula and the definition of $\mathbf{V_u}$ in (2).

In consequence,

$$\frac{1}{2}\frac{\partial \log |\mathbf{V_u}|}{\partial P_{i,j}} - \frac{1}{2}\frac{\partial \log |\rho^2 \mathbf{C}|}{\partial P_{i,j}} = \rho^2\text{tr}\left(\boldsymbol{\delta}_j^{\text{T}}\boldsymbol{\delta}_j\boldsymbol{\delta}_i^{\text{T}}\left(\boldsymbol{\Pi}_{\tilde{\nu}} - \boldsymbol{\Pi}_{\tilde{\nu}}\mathbf{V_u}\boldsymbol{\Pi}_{\tilde{\nu}}\right)\boldsymbol{\Delta}\mathbf{D}\boldsymbol{\delta}_i\boldsymbol{\delta}_i^{\text{T}}\boldsymbol{\Delta}\boldsymbol{\Delta}\boldsymbol{\Delta}\mathbf{P}\boldsymbol{\delta}_j\right)$$

$$+ \rho^2\text{tr}\left(\boldsymbol{\delta}_j^{\text{T}}\boldsymbol{\delta}_j\boldsymbol{\delta}_i^{\text{T}}\mathbf{P}\mathbf{P}^{\text{T}}\boldsymbol{\Delta}\left(\boldsymbol{\Pi}_{\tilde{\nu}} - \boldsymbol{\Pi}_{\tilde{\nu}}\mathbf{V_u}\boldsymbol{\Pi}_{\tilde{\nu}}\right)\boldsymbol{\delta}_i\boldsymbol{\delta}_i^{\text{T}}\boldsymbol{\Delta}\boldsymbol{\Delta}\boldsymbol{\Delta}\mathbf{P}\boldsymbol{\delta}_j\right)$$

$$- \rho^2\text{tr}\left(\boldsymbol{\delta}_j^{\text{T}}\mathbf{P}^{\text{T}}\boldsymbol{\Delta}\left(\boldsymbol{\Pi}_{\tilde{\nu}} - \boldsymbol{\Pi}_{\tilde{\nu}}\mathbf{V_u}\boldsymbol{\Pi}_{\tilde{\nu}}\right)\boldsymbol{\Delta}\boldsymbol{\delta}_i\right)$$

$$= \rho^2\text{tr}\left(\boldsymbol{\delta}_i^{\text{T}}\left(\boldsymbol{\Pi}_{\tilde{\nu}} - \boldsymbol{\Pi}_{\tilde{\nu}}\mathbf{V_u}\boldsymbol{\Pi}_{\tilde{\nu}}\right)\boldsymbol{\Delta}\mathbf{D}\boldsymbol{\delta}_i\boldsymbol{\delta}_i^{\text{T}}\boldsymbol{\Delta}\boldsymbol{\Delta}\boldsymbol{\Delta}\mathbf{P}\boldsymbol{\delta}_j\right)$$

$$+ \rho^2\text{tr}\left(\boldsymbol{\delta}_i^{\text{T}}\mathbf{P}\mathbf{P}^{\text{T}}\boldsymbol{\Delta}\left(\boldsymbol{\Pi}_{\tilde{\nu}} - \boldsymbol{\Pi}_{\tilde{\nu}}\mathbf{V_u}\boldsymbol{\Pi}_{\tilde{\nu}}\right)\boldsymbol{\delta}_i\boldsymbol{\delta}_i^{\text{T}}\boldsymbol{\Delta}\boldsymbol{\Delta}\boldsymbol{\Delta}\mathbf{P}\boldsymbol{\delta}_j\right)$$

$$- \rho^2\text{tr}\left(\boldsymbol{\delta}_j^{\text{T}}\mathbf{P}^{\text{T}}\boldsymbol{\Delta}\left(\boldsymbol{\Pi}_{\tilde{\nu}} - \boldsymbol{\Pi}_{\tilde{\nu}}\mathbf{V_u}\boldsymbol{\Pi}_{\tilde{\nu}}\right)\boldsymbol{\Delta}\boldsymbol{\delta}_i\right) . \tag{47}$$

A similar derivation can be done for the remaining terms. This gives

$$\frac{1}{2}\frac{\partial \log |\mathbf{V_v}|}{\partial P_{i,j}} - \frac{1}{2}\frac{\partial \log |\gamma^2 \mathbf{C}|}{\partial P_{i,j}} = \gamma^2\text{tr}\left(\boldsymbol{\delta}_i^{\text{T}}\left(\boldsymbol{\Pi}_{\tilde{\xi}} - \boldsymbol{\Pi}_{\tilde{\xi}}\mathbf{V_v}\boldsymbol{\Pi}_{\tilde{\xi}}\right)\boldsymbol{\Delta}\mathbf{D}\boldsymbol{\delta}_i\boldsymbol{\delta}_i^{\text{T}}\boldsymbol{\Delta}\boldsymbol{\Delta}\boldsymbol{\Delta}\mathbf{P}\boldsymbol{\delta}_j\right)$$

$$+ \gamma^2\text{tr}\left(\boldsymbol{\delta}_i^{\text{T}}\mathbf{P}\mathbf{P}^{\text{T}}\boldsymbol{\Delta}\left(\boldsymbol{\Pi}_{\tilde{\xi}} - \boldsymbol{\Pi}_{\tilde{\xi}}\mathbf{V_v}\boldsymbol{\Pi}_{\tilde{\xi}}\right)\boldsymbol{\delta}_i\boldsymbol{\delta}_i^{\text{T}}\boldsymbol{\Delta}\boldsymbol{\Delta}\boldsymbol{\Delta}\mathbf{P}\boldsymbol{\delta}_j\right)$$

$$- \gamma^2\text{tr}\left(\boldsymbol{\delta}_j^{\text{T}}\mathbf{P}^{\text{T}}\boldsymbol{\Delta}\left(\boldsymbol{\Pi}_{\tilde{\xi}} - \boldsymbol{\Pi}_{\tilde{\xi}}\mathbf{V_v}\boldsymbol{\Pi}_{\tilde{\xi}}\right)\boldsymbol{\Delta}\boldsymbol{\delta}_i\right) . \tag{48}$$

In summary, the total gradient with respect to $\mathbf{P}$ is:

$$
\begin{aligned}
\frac{\partial \tilde{Z}}{\partial \mathbf{P}} &= \left(\rho^2 \mathbf{H}_{\tilde{\nu}} + \rho^2 \mathbf{F}_{\tilde{\nu}} + \gamma^2 \mathbf{H}_{\tilde{\xi}} + \gamma^2 \mathbf{F}_{\tilde{\xi}}\right) \boldsymbol{\Delta}\boldsymbol{\Delta}\boldsymbol{\Delta}\mathbf{P} - \boldsymbol{\Delta}\left(\rho^2 \boldsymbol{\Pi}_{\tilde{\nu}} - \rho^2 \boldsymbol{\Pi}_{\tilde{\nu}}\mathbf{V}_{\mathbf{u}}\boldsymbol{\Pi}_{\tilde{\nu}} + \gamma^2 \boldsymbol{\Pi}_{\tilde{\xi}} - \gamma^2 \boldsymbol{\Pi}_{\tilde{\xi}}\mathbf{V}_{\mathbf{v}}\boldsymbol{\Pi}_{\tilde{\xi}}\right) \boldsymbol{\Delta}\mathbf{P} \\
&= \left(\rho^2 \mathbf{H}_{\tilde{\nu}} + \rho^2 \mathbf{F}_{\tilde{\nu}} + \gamma^2 \mathbf{H}_{\tilde{\xi}} + \gamma^2 \mathbf{F}_{\tilde{\xi}}\right) \boldsymbol{\Delta}\boldsymbol{\Delta}\boldsymbol{\Delta}\mathbf{P} - \rho^2 \boldsymbol{\Delta}\boldsymbol{\Pi}_{\tilde{\nu}}\boldsymbol{\Delta}\mathbf{P} - \gamma^2 \boldsymbol{\Delta}\boldsymbol{\Pi}_{\tilde{\xi}}\boldsymbol{\Delta}\mathbf{P} \\
&\quad + \rho^2 \boldsymbol{\Delta}\boldsymbol{\Pi}_{\tilde{\nu}}\mathbf{V}_{\mathbf{u}}\boldsymbol{\Pi}_{\tilde{\nu}}\boldsymbol{\Delta}\mathbf{P} + \gamma^2 \boldsymbol{\Delta}\boldsymbol{\Pi}_{\tilde{\xi}}\mathbf{V}_{\mathbf{v}}\boldsymbol{\Pi}_{\tilde{\xi}}\boldsymbol{\Delta}\mathbf{P} \\
&= \left[\left(\rho^2 \mathbf{H}_{\tilde{\nu}} + \rho^2 \mathbf{F}_{\tilde{\nu}} + \gamma^2 \mathbf{H}_{\tilde{\xi}} + \gamma^2 \mathbf{F}_{\tilde{\xi}}\right) \boldsymbol{\Delta}\boldsymbol{\Delta}\boldsymbol{\Delta} - \rho^2 \boldsymbol{\Delta}\boldsymbol{\Pi}_{\tilde{\nu}}\boldsymbol{\Delta} - \gamma^2 \boldsymbol{\Delta}\boldsymbol{\Pi}_{\tilde{\xi}}\boldsymbol{\Delta}\right] \mathbf{P} \\
&\quad + \rho^2 \boldsymbol{\Delta}\boldsymbol{\Pi}_{\tilde{\nu}}\mathbf{V}_{\mathbf{u}}\boldsymbol{\Pi}_{\tilde{\nu}}\boldsymbol{\Delta}\mathbf{P} + \gamma^2 \boldsymbol{\Delta}\boldsymbol{\Pi}_{\tilde{\xi}}\mathbf{V}_{\mathbf{v}}\boldsymbol{\Pi}_{\tilde{\xi}}\boldsymbol{\Delta}\mathbf{P} \\
&= \left[\left(\rho^2 \mathbf{H}_{\tilde{\nu}} + \rho^2 \mathbf{F}_{\tilde{\nu}} + \gamma^2 \mathbf{H}_{\tilde{\xi}} + \gamma^2 \mathbf{F}_{\tilde{\xi}}\right) \boldsymbol{\Delta}\boldsymbol{\Delta}\boldsymbol{\Delta} - \rho^2 \boldsymbol{\Delta}\boldsymbol{\Pi}_{\tilde{\nu}}\boldsymbol{\Delta} - \gamma^2 \boldsymbol{\Delta}\boldsymbol{\Pi}_{\tilde{\xi}}\boldsymbol{\Delta}\right] \mathbf{P} \\
&\quad + \rho^2 \boldsymbol{\Delta}\boldsymbol{\Pi}_{\tilde{\nu}}\mathbf{G}_{\rho}\boldsymbol{\Pi}_{\tilde{\nu}} + \boldsymbol{\Delta}\boldsymbol{\Pi}_{\tilde{\nu}}\mathbf{L}_{\rho}\left(\mathbf{I} + \mathbf{P}^{\mathrm{T}}\mathbf{B}_{\rho}\mathbf{P}\right)^{-1} \mathbf{L}_{\rho}^{\mathrm{T}}\boldsymbol{\Pi}_{\tilde{\nu}}\boldsymbol{\Delta}\mathbf{P} \\
&\quad + \gamma^2 \boldsymbol{\Delta}\boldsymbol{\Pi}_{\tilde{\xi}}\mathbf{G}_{\gamma}\boldsymbol{\Pi}_{\tilde{\xi}} + \gamma^2 \boldsymbol{\Delta}\boldsymbol{\Pi}_{\tilde{\xi}}\mathbf{L}_{\gamma}\left(\mathbf{I} + \mathbf{P}^{\mathrm{T}}\mathbf{B}_{\gamma}\mathbf{P}\right)^{-1} \mathbf{L}_{\gamma}^{\mathrm{T}}\boldsymbol{\Pi}_{\tilde{\xi}}\boldsymbol{\Delta}\mathbf{P} \,,
\end{aligned}
\tag{49}
$$

where we have defined the diagonal matrices

$$
\begin{aligned}
\mathbf{H}_{\tilde{\nu}} &= \mathrm{diag}\left(\left(\boldsymbol{\Pi}_{\tilde{\nu}} - \boldsymbol{\Pi}_{\tilde{\nu}}\mathbf{V}_{\mathbf{u}}\boldsymbol{\Pi}_{\tilde{\nu}}\right) \boldsymbol{\Delta}\mathbf{D}\right) \,, &\quad \mathbf{H}_{\tilde{\xi}} &= \mathrm{diag}\left(\left(\boldsymbol{\Pi}_{\tilde{\xi}} - \boldsymbol{\Pi}_{\tilde{\xi}}\mathbf{V}_{\mathbf{v}}\boldsymbol{\Pi}_{\tilde{\xi}}\right) \boldsymbol{\Delta}\mathbf{D}\right) \,, \\
\mathbf{F}_{\tilde{\nu}} &= \mathrm{diag}\left(\mathbf{P}\mathbf{P}^{\mathrm{T}}\boldsymbol{\Delta}\left(\boldsymbol{\Pi}_{\tilde{\nu}} - \boldsymbol{\Pi}_{\tilde{\nu}}\mathbf{V}_{\mathbf{u}}\boldsymbol{\Pi}_{\tilde{\nu}}\right)\right) \,, &\quad \mathbf{F}_{\tilde{\xi}} &= \mathrm{diag}\left(\mathbf{P}\mathbf{P}^{\mathrm{T}}\boldsymbol{\Delta}\left(\boldsymbol{\Pi}_{\tilde{\xi}} - \boldsymbol{\Pi}_{\tilde{\xi}}\mathbf{V}_{\mathbf{v}}\boldsymbol{\Pi}_{\tilde{\xi}}\right)\right) \,.
\end{aligned}
\tag{50}
$$

Furthermore, the required values can be obtained with cost $\mathcal{O}(n^2 d)$ if $m = n$ and $n \ll d$. For this, the form of $\mathbf{V}_{\mathbf{u}}$ and $\mathbf{V}_{\mathbf{v}}$ given in (10) and (11) has been used. Note also that most matrices are diagonal.

# References

[1] Daniel Hernández-Lobato. *Prediction Based on Averages over Automatically Induced Learners: Ensemble Methods and Bayesian Techniques*. PhD thesis, Computer Science Department, Universidad Autónoma de Madrid, 2009. Online available at: http://arantxa.ii.uam.es/d̃hernan/docs/Thesis_color_links.pdf.