[Reviews · NeurIPS 2013]

Submitted by Assigned_Reviewer_4

Paper Summary:

The paper handles a problem that has garnered growing research interest: Learning sparse solutions for problems with relatively small amount of samples and many features (i.e.; mandate feature selection),
where we try to leverage data from several learning tasks over the same set of features. The assumption is that while the learning tasks are different they may share some characteristics that would enable us to generalize better. Specifically, the authors concentrate on the case where the learning tasks may share a dependency structure between the variables, though whether they are actually "active" and the magnitude of their coefficients (w) in each learning task may differ. The coefficient for each feature in each learning task is controlled via a horseshoe prior to insure sparseness. However, instead of assuming these are independent priors for each feature, the authors introduce a correlation matrix (C) over the hyper parameters (u,v) that control the features coefficients. They assume that matrix (C) is shared between the learning tasks. In this learning setting, we are generally interested in two things: (1) prediction for new data (2) detecting what are the "active" features for each learning task, given by z = (w,u,v) for each task. Since the integrals do not have a closed form solution they resort to approximate inference using EP. For testing their algorithm the authors use two datasets: A synthetic dataset which has a clear regularity in the selected active features in each task (active iff mod(i/16) = j where j in 1...16 and i in 1....128) and a 10x10 digitized handwritten digits. In both cases they show improved results compared to a host of other algorithms that do not utilize mulit-task learning or use multi-task learning but assume the active features are shared or at least have no dependency structure.

Pros:
1. The paper includes comparison to several other methods, and demonstrate improved performance for those tasks.
2. The method developed is new making good use of a whole host of ML related methods. These methods have been gaining interest: Sparse Bayesian priors such as the horseshoe with alternative representations of it, EP with various tricks to make it more efficient. Thus, the paper has the side benefit of educating the readers while the authors claim significant speedup for EP.
3. The intro to the horseshoe prior usage is clearly written.

Cons:
1. While the method is novel and non-trivial the overall relevance of it for real life applications is not clear, and the authors make no attempt at clarifying this. Several things stand out: First, the authors make a general comment that their approach is useful since the different learning tasks "share only the dependency structure" (p.2, top). True, sharing exactly the same active features and coefficients is a stronger assumptions, but other methods relax these assumptions. Sharing (exactly) the same dependency structure actually seems quite a strong assumption, and not necessarily a realistic one if you think of applications such as networks in the bio-medical domain. The authors claim this to be "a very flexible assumption" (p. 8 conclusions) but do not support this in any way. Specifically, they only evaluate their method on two datasets with very clear and unique regularities in their feature dependency structure. Comparison on cases where some features are shared but with no underlying dependency structure exists is warranted. Finally, the authors mention only theoretical computational costs. The experiments involve only small toy data. It is not clear how such a method would perform compared to alternatives when number of samples (n), number of tasks (k) and number of features (f) grows. The latter is especially problematic as the method aimes to reconstruct the correlation matrix between all features.

2. The writeup is missing crucial details for demonstrating mathematical soundness required for publication. The authors constantly use arguments such as "The gradient of .... can be used for this task" (p. 4), "the gradient of.... with respect to .... can be easily computed in terms of the ...." p. 5 Then sate "The derivations are long and tedious and hence omitted here" (p. 6). Similarly, all the actual implementation of the EP in Sec. 3 is only reviewed. The details should be given in a matching supplementary for completeness.

3. The write up is not well balanced/structured. While the main novel element is the estimation of the joint dependency structure the authors spend very little on it. Instead they give a clear but maybe too long intro to the horseshoe prior. Specifically, Figure 1 is taken straight out of Caravlho et al 2009 (Fig 1,2) and the authors should at least point that out clearly and/or retain only say Fig1c. Sec 3 should be extended and and requires supp (see above). In the experiments the author spend almost a full page (p. 7) on a synthetic data which can only be described as a sanity check. The 3 panels in Fig 3 are too much. The set up of the digits experiments is not clear and should be revised (e.g. where does n = 75 comes from? "100" appears as the value of too many different params not defined, which is confusing).

Other:

1. On p. 6 the authors state they define P (which is at the heart of the correlation matrix C) to have m x d params, and set m = n where n is #samples.
But in the multi class learning framework they claim each learning task can have a different number of samples n_k, though P is presumably the same. Can you explain?

2. Fig 2 the annotation "f" for the top factor is missing
Summary: A technically interesting method. The relevance for real life application is not clear, and crucial details are missing.

Submitted by Assigned_Reviewer_5

The paper proposes a novel probabilistic model for
sparse models in multitask learning.

As I am not familiar with probabilistic learning models and the related
literature, rhis review is thus an educated guess on the
work proposed by the authors.

However, it seems to me that the idea of integrating a term that
controls the variable dependency in the feature selection
process is novel and interesting.

Experimental results how good the model performs on some toy
datasets as well as on handwritten digits.

My only concern about the work would be that only one "real"
dataset has been considered in the experiments and showing
the perfomance of the model on more datasets would have strenghten
the paper.
Summary: Novel probabilistic model for feature selection in multi-task learning. The main
contribution consists in adding a correlation term that can be learned.


Submitted by Assigned_Reviewer_7

The paper presents a method for multi-task learning that allows for learning the dependencies between features for each task. Learning dependencies between features is possible through a modification of the horseshoe prior, introduced by the authors. Extension to multi-task learning is allowed by letting each task share the same covariance structure involved in the construction of the modified horseshoe prior. The method is applied to a synthetic dataset and a real-world dataset.

A clear contribution of the paper is that the authors have found a clever way to learn dependencies between features by a minor modification to the horseshoe prior. The extension to multi-task learning is straightforward.

One important part that is missing in the paper is the related work that has been done for some years now on Multi-task Feature Selection. In that piece of literature there are several methods that allow for feature selection in multi-task scenarios, and there are different variants of those methods, which also allow for learning different groups of features according to each task. I recommend the authors to have a look at the related work section of the paper “Exclusive Lasso for Multi-task Feature Selection” by Y Zhou, et al, AISTATS 2010. Including results using any of those methods would clearly make a stronger paper.

I also find the real-data example on the MNIST dataset very artificial. I wonder if there is not a real-world data example where you can apply the method without having to modify the original dataset or manipulate the original dataset in any way.
Summary: A clever way to introduce dependencies between features by means of the horseshoe prior. The authors ignore an important piece of the literature, and in that sense the method may lose originality.
Author Feedback

Author rebuttal: We thank the reviewers for their interesting comments and careful analysis.

We disagree with the statement that our submitted work is incremental and unlikely to have much impact. Introducing dependencies in the feature selection process dramatically improves the induction process under the sparsity assumption (see [7,8,10]). For example, an extreme case of these dependencies appears in the group LASSO which produces significant improvements with respect to the LASSO [10]. However, most times the dependency structure, or similarly, the groups in the group LASSO, must be given by some expert. Finding an automatic and general method to obtain this information is an open problem. Our work shows how this information (e.g. groups in the group LASSO) can be automatically inferred from the data. See for example Figure 3, left bottom. Each black square in the correlation matrix represents a learnt group of jointly relevant or irrelevant features. We believe this is a problem whose solution will receive interest from the community.

Potential applications of the method include problems where induction under the sparsity assumption can be beneficial, e.g., analysis of micro-array data, compressed sensing, gene-network identification, analysis of fMRI data, source localization or image denoiseing. For example, natural images have sparse representations in a wavelet basis (the images employed in Section 4.2 are already sparse). A noisy image can be decomposed in multiple patches and the wavelet coefficients for each patch can be induced using the proposed method. Each patch will correspond to a learning task. The original image can be then reconstructed from the resulting coefficients.

Specific comments for each reviewer:

-Reviewer_4

Our intention was to express that assuming a common dependency structure is less restrictive than assuming shared relevant and irrelevant features.

We mention in our manuscript (and compare with) other methods that relax the assumption of shared relevant and irrelevant features across tasks, e.g. the dirty model (DM) in [14].

Shared relevant and irrelevant features among tasks already introduces dependencies in the feature selection process. In each task there are two groups. One group of features expected to be jointly selected (the relevant ones) and one group of features expected to be non-selected (the irrelevant ones). Assuming shared dependencies will probably perform better than single-task learning. However, assuming shared relevant and irrelevant features is expected to perform best since it is optimal.

The cost of the method is O(Kn^2d), with K the number of tasks, n the number of samples of each task and d the number of features. This is valid if m in (5) is O(n). The total cost is hence linear in d. Increasing m is justified if very complex dependency patters have to be learnt. If m=d no restriction at all is put in C, but the cost is O(Kd^3). Setting m=n < d servers as a regularizer for C, leads to good results in our experiments and has computational advantages.

Implementing the EP algorithm and the gradient computation is tedious but mechanic. The missing details will be given in the supplementary material. The source code of the method will also be publicly released.

Figure 1 has been included to motivate the good properties of the horseshoe prior for sparsity induction and because not all readers are familiar with this prior. This figure is based on the one displayed in [4], but has been generated by us.

In Section 4.2 setting n=75 and d = K = 100 is consistent with n < d and gives good discrimination results among the methods evaluated.

It is possible to have a different number of samples n_k for each task (not in DM). In that case one would like to set m = min n_k for k=1,...,K so that the computational cost is O(sum_k=1^K n_k^2 d).

-Reviewer_7:

We thank the reviewer for the references mentioned by Y. Zhou et al. These references will be included in the paper. [13] is already included.

Most of the references (Xiong et al., Argyriou et al., Obozinski et al. and Jebara) assume jointly relevant and irrelevant features across tasks. This agrees with our statement in the introduction saying that traditionally, methods for multi-task learning under the sparsity assumption have focused on this hypothesis. Thus, those methods are not expected to perform, in the problems analyzed, significantly better than HS_MT, the method we compare with and which explicitly makes that hypothesis.

The methods of Argyriou et al. and Obozinski et al. are particular cases of the group LASSO [10]. The method we compare with, DM, includes the group LASSO as a particular case. Thus, those methods are not expected to perform better than DM.

Lee et al. consider tasks that have non-overlapping features or where feature relevance can vary. However, they strongly rely on having meta-feature information available for this. Their method cannot be applied in situations where this information is not available, as it is our case, without being simply reduced to the group LASSO or the LASSO.

The work by Chan et al. does not incorporate the sparsity assumption in the induction process. Thus, it is not expected to be competitive in the problems analyzed (which are sparse) with the methods that do incorporate this assumption (HS_Dep, HS_ST, HS_MT and DM).

The references indicated by the reviewer, although related to the problem of multi-task feature selection and relevant for the present work, they are not directly related to the problem of learning dependencies in the feature selection process, which is the main topic of the submitted paper.

It is not accurate to say that we do not compare with other related methods for multi-task feature selection. We compare results with HT_MT and DM, which are two approaches proposed in the literature for this purpose. Furthermore, DM includes the group LASSO as a particular case.